# Characterization of protein unfolding by fast cross-linking mass spectrometry using di-*ortho*-phthalaldehyde cross-linkers

Jian-Hua Wang [1,2,9], Yu-Liang Tang[3,9], Zhou Gong[4], Rohit Jain [5], Fan Xiao[3], Yu Zhou [1,2], Dan Tan[1], Qiang Li[3], Niu Huang [1,2], Shu-Qun Liu[6], Keqiong Ye [7,8], Chun Tang [3,4✉], Meng-Qiu Dong [1,2✉] & Xiaoguang Lei [3✉]

Chemical cross-linking of proteins coupled with mass spectrometry is widely used in protein structural analysis. In this study we develop a class of non-hydrolyzable amine-selective di-*ortho*-phthalaldehyde (DOPA) cross-linkers, one of which is called DOPA2. Cross-linking of proteins with DOPA2 is 60–120 times faster than that with the N-hydroxysuccinimide ester cross-linker DSS. Compared with DSS cross-links, DOPA2 cross-links show better agreement with the crystal structures of tested proteins. More importantly, DOPA2 has unique advantages when working at low pH, low temperature, or in the presence of denaturants. Using staphylococcal nuclease, bovine serum albumin, and bovine pancreatic ribonuclease A, we demonstrate that DOPA2 cross-linking provides abundant spatial information about the conformations of progressively denatured forms of these proteins. Furthermore, DOPA2 cross-linking allows time-course analysis of protein conformational changes during denaturant-induced unfolding.

[1] National Institute of Biological Sciences (NIBS), 102206 Beijing, China. [2] Tsinghua Institute of Multidisciplinary Biomedical Research, Tsinghua University, 102206 Beijing, China. [3] Beijing National Laboratory for Molecular Sciences, Key Laboratory of Bioorganic Chemistry and Molecular Engineering of Ministry of Education, Synthetic and Functional Biomolecules Center, College of Chemistry and Molecular Engineering, Peking-Tsinghua Center for Life Science, Peking University, 100871 Beijing, China. [4] Innovation Academy for Precision Measurement Science and Technology, Chinese Academy of Sciences, 430071 Wuhan, Hubei, China. [5] University of Massachusetts Medical School, Worcester 01605, USA. [6] State Key Laboratory for Conservation and Utilization of Bio-Resources in Yunnan, Yunnan University, 650091 Kunming, Yunnan, China. [7] Key Laboratory of RNA Biology, CAS Center for Excellence in Biomacromolecules, Institute of Biophysics, Chinese Academy of Sciences, 100101 Beijing, China. [8] University of Chinese Academy of Sciences, 100049 Beijing, China. [9] These authors contributed equally: Jian-Hua Wang, Yu-Liang Tang. ✉email: Tang_Chun@pku.edu.cn; dongmengqiu@nibs.ac.cn; xglei@pku.edu.cn

Proteins constantly undergo conformational changes. Protein motions span a wide range of temporal and spatial scales, from the fast fluctuations of the amino acid side chains in picoseconds to the moderate fluctuations of the surface loops in nanoseconds and to the slow collective motions of the domain/ entire protein in microseconds to seconds[1,2]. The biological function of a protein is rooted in its physical motions and dynamic properties[1,3]. Proteins experience the most dramatic conformational changes during folding, unfolding, and refolding, all of which occur over a wide time-scale from subseconds to minutes[4,5]. Characterizing the different conformational states of native folded and denatured unfolded proteins, as well as related conformational transitions is of fundamental importance to biology and of practical value to drug development.

A variety of tools have been developed for conformational studies of proteins. X-ray crystallography and single-particle cryo-electron microscopy (cryo-EM) have revolutionized structural biology by taking "snapshots" of proteins or protein complexes at atomic resolution[6,7]. However, these technologies are less effective for revealing protein dynamics under physiological conditions. NMR spectroscopy can provide valuable information on the dynamics of a protein, but is typically limited to proteins of <50 kDa[8,9]. Fluorescence resonance energy transfer (FRET)[10,11] and electron paramagnetic resonance (EPR)[12] circumvent protein size limitations by using targeted insertion of donor and acceptor fluorophores, or by using two spin labels, respectively. However, these require a prior hypothesis and elaborate protein manipulation. Small-angle X-ray scattering (SAXS) is increasingly used to characterize protein dynamics in solution without the use of fluorophores[13,14]. However, it falls short in elucidating protein conformational change to a residue-specific level[15].

Mass spectrometry-based methods (such as hydrogen-deuterium exchange[16,17] and hydroxyl radical footprinting[18]) can probe the dynamics of a protein based on the difference in the solvent accessibility among various parts of the protein structure. However, they do not directly provide a three-dimensional (3D) structure. Chemical cross-linking of proteins coupled with mass spectrometry analysis (CXMS, XL-MS, or CLMS) is a straightforward approach for investigating protein structures and protein-protein interactions[19–24]. CXMS has a great potential of capturing the conformational changes in proteins, owing to the rapid linkage of two amino acid residues spatially close to each other. Encouragingly, CXMS has been utilized successfully to visualize the co-existing conformations of a protein[25] or protein complex[26] and to compare the different conformational states of a protein[27] or protein complex[28]. However, to our knowledge, CXMS has not been used to investigate continuous protein conformational changes. This is mostly because cross-linking reactions are usually too slow to occur.

Although a number of chemical cross-linkers can target Lys[29–32], Arg[33], Cys[34], or acidic amino acids[35,36], CXMS has largely focused on the lysine-targeting N-hydroxysuccinimide (NHS) ester cross-linkers such as disuccinimidylsuberate (DSS), bis(sulfosuccinimidyl)suberate (BS³), and disuccinimidyl sulfoxide (DSSO) owing to their high reactivity and an abundance of lysine residues on protein surface[20]. However, the NHS ester-based cross-linking reaction is slow and typically takes 30–60 min on protein substrates[37]. Besides, NHS ester is susceptible to rapid hydrolysis in aqueous solutions. The half-life of an NHS ester is about tens of minutes under the typical reaction conditions[38].

On the other end of the cross-linker spectrum are a variety of photo-cross-linkers, which can react with proteins in seconds, theoretically, upon UV activation[39]. However, because their target sites could be any amino acids, analysis of photo-cross-linking data is faced with a big problem of keeping out false identifications. As such, the development of a lysine-specific cross-linker

that is faster and more stable than NHS ester cross-linkers is an attractive approach to accessing new possibilities of CXMS.

Ortho-phthalaldehyde (OPA) represents an alternative reagent to modify amino groups. The earliest report of OPA reacting with an amino group to form an isoindolinone skeleton can be dated back to 1909[40]. Since then, many OPA-related applications have been reported, including determination of serum protein concentration[41], polymer synthesis[42], and disinfection of surgical equipments[43]. A recent study reported that OPA can selectively modify amino groups on natural protein surfaces[44].

Attracted by the unique properties of OPA—amine-selectivity, fast reaction, and no hydrolysis—we developed a class of amine-selective non-hydrolyzable di-ortho-phthalaldehyde (DOPA) cross-linkers. We show that DOPA2, with a spacer arm of two ethylene glycol units, cross-links proteins ~60 times faster than DSS. Furthermore, we show that DOPA2 can cross-link proteins under extreme conditions such as low temperature, low pH, or in the presence of denaturants. These unique properties of DOPA2 make it an ideal cross-linker for probing protein folding/unfolding intermediate states and for capturing the sequential changes during protein unfolding, as demonstrated with three model proteins staphylococcal nuclease (SNase), bovine serum albumin (BSA), and bovine pancreatic ribonuclease A (RNase A).

## Results

**Development of a class of amine-reactive cross-linkers.** First, we verified that OPA reacts selectively with lysine or peptide N-terminal amine groups by testing it on ten synthetic peptides covering all 20 common amino acids (Supplementary Table 1). Liquid chromatography coupled with mass spectrometry (LCMS) analysis confirmed that the major products were N-substituted phthalimidines[44] formed on lysine $\varepsilon$-NH$_2$ or N-terminal $\alpha$-NH$_2$ ($\Delta$ mass = +116.0262 Da) (See product 1 in Supplementary Table 2 and Supplementary Data 1). The loop-linked side product (product 2, $\Delta$ mass = +98.0156 Da) resulted from the conjugation of OPA with an amino group and with another nucleophilic group on the same peptide; such nucleophilic groups included $\alpha$-NH$_2$ from the free N-terminus, $\varepsilon$-NH$_2$ from lysine, -SH from cysteine, and a phenolic hydroxyl group from tyrosine (Supplementary Table 2 and Supplementary Data 1). The results not only corroborated previous reports[44,45], and established a foundation for further development of OPA-based cross-linkers.

Next, we synthesized three di-ortho-phthalaldehyde (DOPA) cross-linkers (Fig. 1). In DOPA-C$_2$, two OPA moieties are connected via one ethylene group. In DOPA1 and DOPA2, the spacer arm consists of one and two ethylene glycol units, respectively.

Using BSA as a model protein, we optimized the cross-linking conditions for the DOPA cross-linkers (Supplementary Fig. 1). The highest number of cross-linked peptide pairs (referred to as cross-links for short) was obtained after a 10-min reaction at 16:1 (BSA:DOPA, w/w), 25 °C, pH 7.4. Three buffer systems free of primary amines (HEPES, PBS, and trimethylamine) worked similarly well (Supplementary Fig. 1a–c). Cross-linking at a protein:DOPA2 ratio of 4:1 (w/w) also performed well, provided that two parallel protease digestions (trypsin plus Asp-N and trypsin alone) were carried out (Supplementary Fig. 1d). These experiments also allowed us to identify a critical factor for successful DOPA cross-linking— pre-dilution of DOPA to two times its final working concentration and then mixing this 2× working solution with an equal volume of protein solution (Supplementary Fig. 1e).

**DOPA2 cross-linking of proteins is at least 60 times faster than DSS.** Following a previous study[44], we measured the second-order reaction rate constants of OPA and an NHS ester (NHS hereafter) on a free lysine substrate by quenching the reactions at

**Fig. 1 Syntheses of di-*ortho*-phthalaldehyde (DOPA) cross-linkers.** Reagents and conditions: **a** $Me_2Zn$, $RhCl(PPh_3)_3$, THF, 24 h, 76%. **b** DIBAL-H, THF, 12 h. **c** DMP, DCM, 12 h, 98% for two steps. **d** $Cs_2CO_3$, *bis*(2-iodoethyl) ether or 1,2-dibromoethane, ACN, 80 °C, 12 h, $n = 2$, 77%; $n = 1$, 37%. **e** DIBAL-H, THF, 12 h. **f** DMP, DCM, 12 h, $n = 2$, 93% for two steps; $n = 1$, 80% for two steps. DIBAL-H = diisobutylaluminum hydride, DMP = Dess-Martin periodinane.

different time points and analyzing the products by LCMS (Supplementary Fig. 2). As shown, the lysine/NHS reaction had a second-order rate constant of $0.36\,M^{-1}\,s^{-1}$ (Supplementary Fig. 2a, b, e). The lysine/OPA reaction appeared to have an initial phase that was too fast to be measured by LCMS; the second phase of the reaction had a rate constant of $2.24\,M^{-1}\,s^{-1}$ (Supplementary Fig. 2c–e). These rate constants are similar to the reported values[44].

To determine the pseudo-first-order reaction rate constant for OPA, which we found too fast to measure by LCMS, we employed a high-sensitivity FRET assay to monitor in real time the reaction of Cy5-labeled OPA or a Cy5-labeled NHS ester with a Cy3-labeled lysine-containing peptide Cy3-K in tenfold molar excess (Fig. 2a-c, Supplementary Fig. 3). Assuming 100% energy transfer efficiency, we calculated the pseudo-first-order reaction rate constant of Cy5-OPA to be $3.18 \times 10^{-4}\,s^{-1}$ and that of Cy5-NHS to be $9.10 \times 10^{-5}\,s^{-1}$ with the initial concentration of Cy5-OPA or Cy5-NHS ester at $2.5\,\mu M$ and that of Cy3-K at $25\,\mu M$ (Supplementary Fig. 3g, i). The actual values should be higher.

Lastly, we compared the cross-linking reaction rates of DOPA and DSS on proteins (Fig. 2d–g). As shown by SDS-PAGE, 10 s of DOPA2 treatment and 10–20 min of DSS treatment produced similar amounts of covalently linked BSA dimers (Fig. 2d). By LCMS analysis, ~2000 MS2 spectra of cross-linked peptide pairs of BSA were detected after either 10–20 s of DOPA2 reaction or 20 min of DSS reaction (Fig. 2e). For a more rigorous comparison, we purified a recombinant calcium sensor protein cameleon, which contains concatenated calmodulin and the M13 peptide in the middle, flanked by cyan fluorescent protein (CFP) and yellow fluorescent protein (YFP) (Fig. 2f)[46]. Upon binding to $Ca^{2+}$, calmodulin undergoes a conformational change and binds to M13; this in turn brings CFP and YFP to proximity and enables FRET[46]. We reasoned that chemical cross-linking in the presence of calcium would lock CFP and YFP in close distance, leading to persistent FRET signals even after calcium is chelated by EGTA. Indeed, both DOPA2 and DSS cross-linking produced persistent calcium-independent FRET signals, but it took DSS at least 20 min to reach the signal level produced by DOPA2 in 10 s (Fig. 2g and Supplementary Fig. 4).

Figure 2h shows a summary of the above results. On free lysine or peptide substrates, OPA reacts with amino groups 3.5–7.7 times faster than the NHS ester. However, on protein substrates, DOPA2 cross-linking reactions are 60–120 times faster than DSS reactions. In other words, the difference in reaction rate between OPA and NHS ester on amino acids or peptides is augmented when it comes to DOPA2 versus DSS on protein cross-linking. Such a difference suggests that on protein substrates, the kinetics of producing a pair of cross-linked residues are not simple additions of two consecutive OPA or NHS ester reactions.

**Performance of DOPA vs DSS on model proteins.** Using a ten-protein mixture as a test sample, we found that the number of peptide pairs cross-linked by DOPA2 (161) outnumbered those by DOPA1 (117) or DOPA-$C_2$ (57), and two thirds or more of the DOPA1 or DOPA-$C_2$ cross-linked lysine pairs were also found among the DOPA2 dataset (Fig. 3a-c and Supplementary Table 3a). Next, we conducted a side-by-side comparison of DOPA2 and the widely used NHS ester cross-linker DSS on a panel of six model proteins. Although DSS cross-links outnumbered DOPA2 cross-links for five out of six proteins (Fig. 3d), a closer examination of the distance between each pair of cross-linked residues revealed that DOPA cross-links had a higher degree of agreement with the crystal structures of the proteins than DSS cross-links (Fig. 3e and Supplementary Table 3b). For example, by Euclidean distance, 44.8% of the DSS cross-links exceeded the maximum allowed cross-linking distance (24.0 Å for DSS, Cα-Cα), indicating a structural compatibility rate of 55.2% for DSS cross-links. This is much lower than that of DOPA-$C_2$ (78.99%) or DOPA2 cross-links (88.3%) within the maximum allowed cross-linking distance of 24.9 or 30.2 Å, respectively. The same conclusion can be reached using the solvent accessible surface distance (SASD) (Fig. 3e). Of note, the maximum allowed cross-linking distance of DOPA-$C_2$ (24.9 Å) and that of DSS (24.0 Å) are similar, therefore the length of a cross-linker is unlikely the only determinant of structural compatibility; other chemical properties such as flexibility, hydrophobicity, and bulkiness of the cross-linker likely play a role.

We further evaluated DOPA2 against an NHS ester type lysine-lysine cross-linker BSMEG, a zero-length cross-linker EDC, which ligates a lysine residue to a neighboring carboxyl amino acid, and two heterobifunctional photo-cross-linkers SDA and sulfo-LC-SDA (Supplementary Fig. 5). As shown, more cross-linked peptide pairs were identified with DOPA2 than with the other cross-linkers, except for EDC on BSA. On a mixture of ten proteins, though, the highest number of cross-links were identified using DOPA2.

**Advantages of DOPA cross-linking.** Although DOPA2 did not outperform DSS in terms of the number of cross-links identified in conventional cross-linking experiments (Fig. 3d), it did exhibit

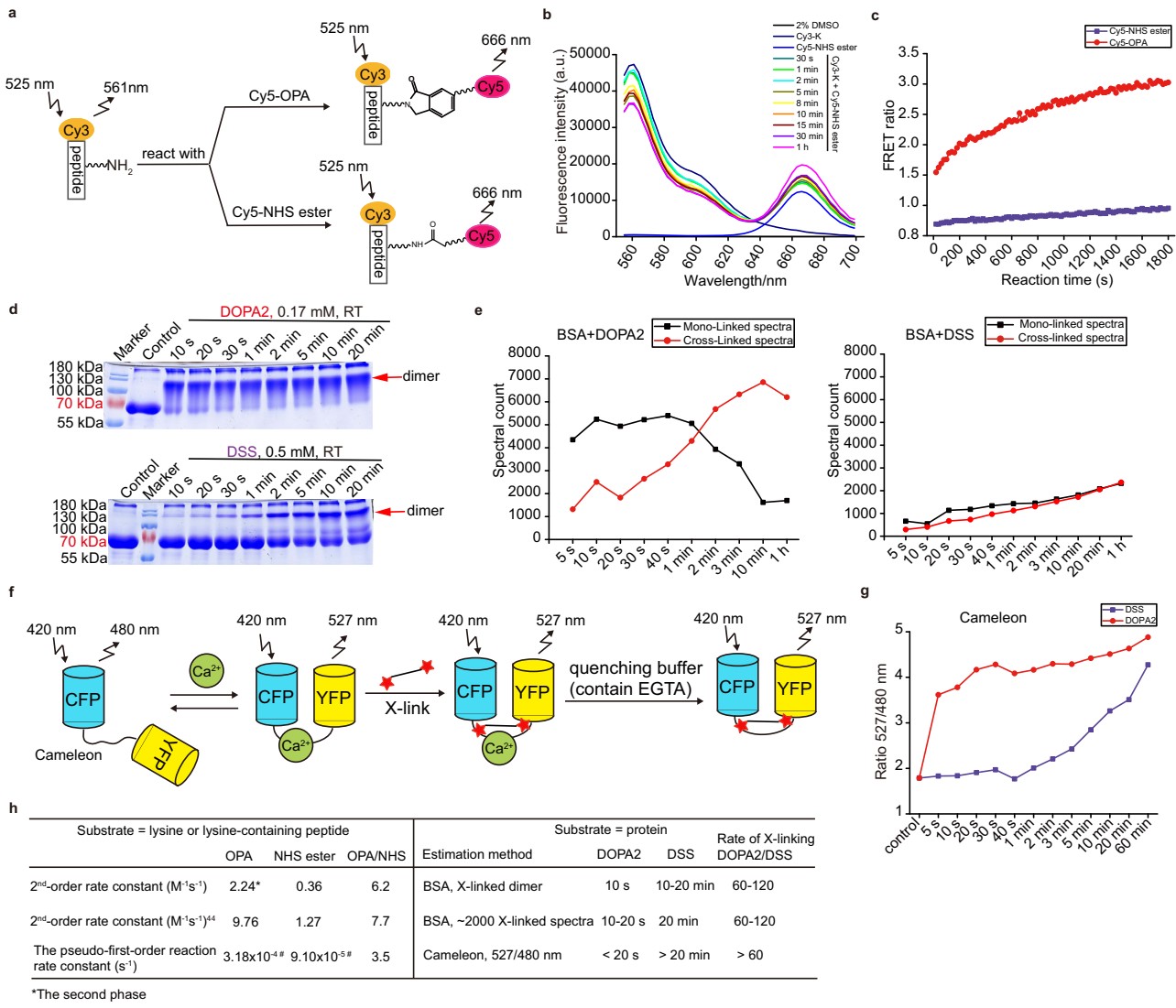

**Fig. 2 DOPA2 is a faster cross-linker than DSS. a** FRET-based workflow of Cy3-K peptide reacting with Cy5-OPA or Cy5-NHS ester. **b** An example of the FRET signal change as the reaction between Cy3-K (12.5 μM) and Cy5-NHS ester (25 μM) proceeds. In the y-axis, a. u. stands for arbitrary units. **c** FRET ratio as a function of reaction time. The reaction was initiated by mixing 25 μM Cy3-K peptide with 2.5 μM Cy5-OPA or Cy5-NHS ester. **d** SDS-PAGE analysis of DOPA2 or DSS cross-linked BSA under different reaction times at room temperature. The similar result was obtained from two more independent experiments. **e** The number of DOPA2 or DSS mono-linked spectra and cross-linked spectra on BSA with increasing reaction time. **f** FRET-based workflow of cross-linking cameleon calcium sensor protein (YC3.6) using DOPA2 or DSS. (CFP, cyan fluorescent protein; YFP, yellow fluorescent protein). **g** The emission fluorescence ratio of 527 and 480 nm with the increasing reaction time. **h** A summary for comparing the reaction rate of OPA and NHS ester, or DOPA2 and DSS. Source data for (**b–d**, **e**, **g**) are provided as a Source Data file.

a unique advantage in cross-linking reactions conducted at low temperature, low pH, or in the presence of denaturants, as detailed below.

We have shown that at room temperature, protein cross-linking is 60–120 times faster with DOPA2 than with DSS (Fig. 2d–g). Even at 0 °C, DOPA2 readily cross-linked BSA within 30 s, whereas DSS hardly generated any BSA dimer bands at 0 °C after 30 min (Fig. 4a). Upon decreasing the pH from 7.4 to 6.0, DOPA2 rather than DSS successfully cross-linked BSA in 10 s (Fig. 4b). A further decrease in the reaction pH to 3.0 completely abolished cross-linking of BSA by either DSS or DOPA2 (Fig. 4b). However, DOPA2 performed similarly well on peptide substrates at pH 3.0 or 7.4 (Fig. 4c and Supplementary Table 4a, b). Therefore, the disappearance of DOPA2 cross-linked BSA dimers at pH 3.0 is more likely a consequence of acid-induced dissociation of BSA dimers. Additionally, we found that DOPA2

but not DSS was able to cross-link proteins in the presence of urea or GdnHCl within 10 s (Fig. 4d, e). Low-level DSS cross-linking was observed in the presence of urea or GdnHCl, but it took much longer, e.g., 15 min in 1–4 M urea or 6 min in 1 M GdnHCl (Fig. 4e). Again, under high concentrations of denaturants (>4 M urea or >1 M GdnHCl), the disappearance of cross-linked BSA dimer bands was at least partly attributable to dissociation of BSA dimers, because DOPA2 reacted with peptides in 6 M GdnHCl (Fig. 4f and Supplementary Table 4c). The above results demonstrate that DOPA2 rapidly cross-link proteins, even under harsh conditions.

Fast cross-linking can be used to capture dynamic changes of proteins, providing that the reaction is stopped promptly. We therefore screened a panel of compounds (ammonia, methylamine, hydrazine, methoxyammonium chloride, and o-benzylhydroxylamine) and found that hydrazine can quench OPA in 5 s

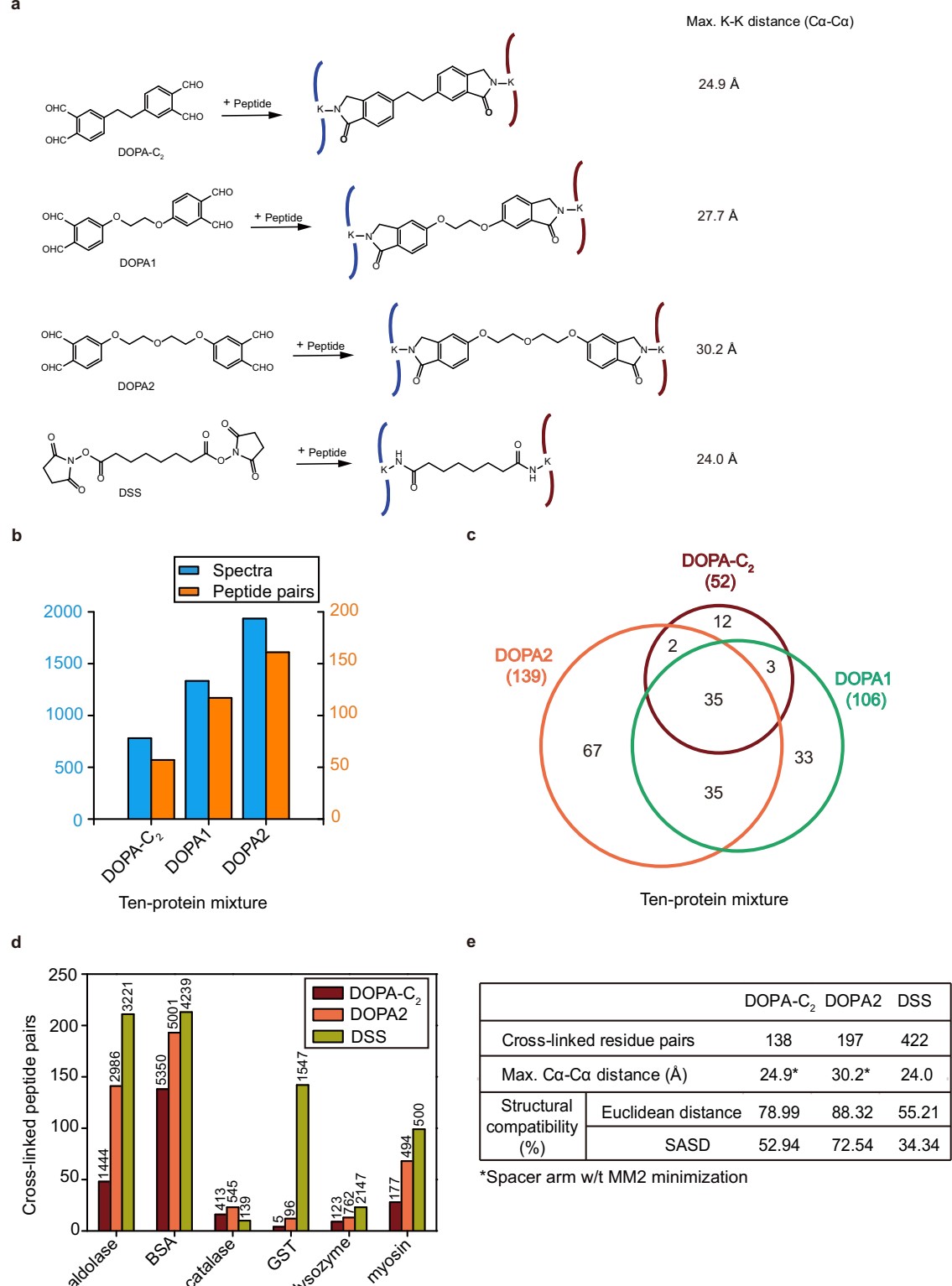

(Supplementary Fig. 6). It is possible that hydrazine may quench the reaction in <5 s, but with manual pipetting we were unable to test shorter time points.

Compared to a 10-min cross-linking reaction, a 10-s reduced the number of DOPA2 cross-link spectra by about two-thirds on the BSA sample (Fig. 2e). More detailed analyses of BSA and a heterodimeric protein complex PUD-1/2 revealed that shortening the cross-linking time from 10 min to 10 s decreased the number

of identified DOPA2-linked residue pairs by 47–59% and that of DSS-linked residue pairs by 68–100% (Supplementary Fig. 7a–d and g–j). In 10-s reactions, 47 DOPA2- vs 14 DSS-linked residue pairs were identified from BSA, and 25 DOPA2- vs zero DSS-linked residue pair were identified from PUD-1/2. The DOPA2 cross-links identified from 10-s reactions fit perfectly with the crystal structure of BSA or PUD-1/2 (Supplementary Fig. 7b, d). For both cross-linkers, the identified cross-links from shorter

**Fig. 3 Evaluating the performance of DOPA. a** Cross-linking products of DOPA-$C_2$, DOPA1, DOPA2, and DSS with peptides. **b** Cross-links identified from a ten-protein mixture using DOPA-$C_2$, DOPA1, and DOPA2. The number of cross-linked spectra is plotted with blue columns, and the number of cross-linked peptide pairs is plotted with orange columns. **c** Venn diagram showing the overlap of residue pairs produced by DOPA-$C_2$, DOPA1, and DOPA2 from the ten-protein mixture. Identified cross-links were filtered by requiring an FDR < 0.01 at the spectra level. **d** Performance of DOPA-$C_2$, DOPA2, and DSS on model proteins. Numbers of cross-linked peptide pairs are indicated with the colored columns, and spectra identified from each sample are shown above the columns. Two independent cross-linking experiments were performed for each protein sample, and analyzed by LC-MS/MS. Identified cross-links were filtered by requiring a FDR < 0.01 at the spectra level. **e** The table displays the percentage of residue pairs that are consistent with the structures of the model proteins, calculated by the use of the Euclidean distance or the solvent accessible surface distance. It also gives the maximum distance restraints and the number of cross-links belonging to each linker. Identified cross-linking residue pairs were filtered by requiring FDR < 0.01 at the spectra level and with spectral counts ≥ 3. Source data for b-e are provided as a Source Data file.

(≤1 min) reactions are a subset, or nearly a subset, of those from longer (≥1 min) reactions (Supplementary Fig. 7e, f and k, l).

We envisaged that such fast, 10-s cross-linking by DOPA2 in the presence or absence of chaotropic denaturants followed by immediate quenching upon the addition of hydrazine would open up new possibilities for CXMS. As such, we explored in the following experiments.

**Analyzing the unfolded states of SNase by DOPA2 cross-linking.** Taking advantage of its unique properties, we used DOPA2 to probe the unfolded states of staphylococcal nuclease (SNase), a model protein for protein folding studies. It has no disulfide bonds and undergoes reversible and efficient folding and unfolding. SNase adopts a globular structure (149 residues) composed of a β-barrel (β-subdomain) and three α-helices (α-subdomain) (Fig. 5a). It is generally accepted that SNase behaves like a cooperative unit and unfolds by two-state transition in equilibrium denaturing reactions. However, an intermediate with a folded β-subdomain and hardly any α-helices can be observed when the folding pathway was perturbed by solvent condition or protein mutations[47,48]. The β-subdomain may also fold first in kinetic folding reactions[49].

The conformation of SNase was first assessed by fluorescence at 325 nm, which primarily came from the single tryptophan residue at position 140, and by circular dichroism (CD) at 222 nm, which reported the amount of α-helices (Fig. 5b, c). Both fluorescence and CD signals showed sigmoidal curves with the transition at 2–4 M urea, indicating of a cooperative protein unfolding as expected[48].

Next, we probed the conformational change of SNase with DOPA2-cross-linking in urea-induced equilibrium denaturation reactions. In detail, equal amounts of SNase were incubated at room temperature in 0–8 M urea to equilibrium, followed by DOPA2 treatment for 10 s, protease digestion, and LCMS analysis (Fig. 5d). We made sure that only the most reliable cross-links were used in the subsequent analysis by applying a stringent cutoff (FDR < 0.01, at least four MS2 spectra at E-value < $1 \times 10^{-8}$) to the pLink 2[50] search results.

Based on how their abundances changed from zero to 8 M urea, the cross-linked residue pairs were clustered into three groups (Fig. 5e). The numbers of matched MS2 spectra for each pair of cross-linked sites were first normalized within a sample and then normalized across the six conditions (shown as Z-scores on the left axis). In the CD data, the mean residue ellipticity (MRE) at 222 nm is an indicator of secondary structure content, which we extracted from Fig. 5c and plotted against the urea concentration in Fig. 5e (gray line, y-axis values on the right). The abundance of Cluster A cross-links was significantly reduced across the transition zone, whereas Cluster B cross-links were increased. With respect to the Cα- Cα distances of cross-linked residue pairs, those of Cluster A were largely consistent with the crystal structure of SNase (Fig. 5f), whereas those of Cluster B were not, suggesting that they originated from the folded and

unfolded state, respectively. Hence, the abundance changes of Cluster A and B cross-links recapitulate the two-state unfolding process of SNase.

Interestingly, DOPA2 cross-linking additionally revealed conformational changes not detectable by either CD or fluorescence. A few cross-links, known as Cluster C, were abruptly reduced at 1 M urea before the major transition occurred. All these cross-links are within the α-subdomain, and they all involve α-helix 3, suggesting that α-helix 3 is unpacked before the breakdown of the entire structure. Thus, the rapid cross-linking by DOPA2 provides a sensitive mean to monitor both global and local conformational changes in protein unfolding reactions, and paints a more complicated picture of SNase unfolding.

**Analyzing the unfolded states of BSA by DOPA2 cross-linking.** Using the same method, we analyzed denaturation of BSA. This 583-aa protein is much larger than SNase and exists in a monomer-dimer equilibrium in solution. Each BSA monomer has three helical domains arranged in the shape of a heart: Domain I (1–172 aa) and Domain III (373–583 aa) are the left and the right atrium, respectively, joined by Domain II (173–372 aa), the left and right ventricles in one[51].

Previous studies using a variety of techniques have shown that BSA follows a two-stage, three-state unfolding route going from zero to 8 M urea[52–55]. For example, both the CD signal in the far-UV region and the fluorescence intensity change of tryptophan residues, primarily Trp214 (Trp213 in Fig. 6), report no or little change of BSA in 1 and 2 M urea, followed by a gradual increase or decrease up to 7 M urea, and no further change in 8 M urea[55]. Though appearing to be a two-state transition, the tryptophan fluorescence exhibits blue shift going from zero to 4 M urea, and a red shift going from 5 to 8 M urea[55], which indicates the presence of an intermediate unfolding state in 4–5 M urea.

In this study, a total of 87 high-quality cross-links identified from the BSA samples in 0–8 M urea (Fig. 6) characterized BSA denaturation in greater detail. As shown in Fig. 6a and Supplementary Fig. 8, the Lys524-Lys524 cross-link was the most sensitive one to perturbation by urea. It is the only intermolecular cross-link identified unambiguously in this experiment. Its abundance nearly halved upon 1 M urea and dropped all the way down upon 2 M urea.

The 16 cross-linked lysine pairs of Cluster 1 were the second most sensitive to urea (Fig. 6b). The two-linked residues in each pair tend to be far apart in the primary sequence (69% >100 aa, 38% >200 aa). In fact, seven of them are between-domain cross-links. In addition, we observed ten cross-links that bridge across the central cleft of the heart-shaped BSA monomer where there is a dearth of hydrogen bonds. The lack of a force to hold the cleft in position may explain why the cross-cleft cross-linking is easily disrupted by urea.

Less sensitive to urea were the cross-links of Cluster 2 (Fig. 6c). Cluster 1 and 2 both diminished in abundance as the urea concentration increased. However, unlike Cluster 1, Cluster 2

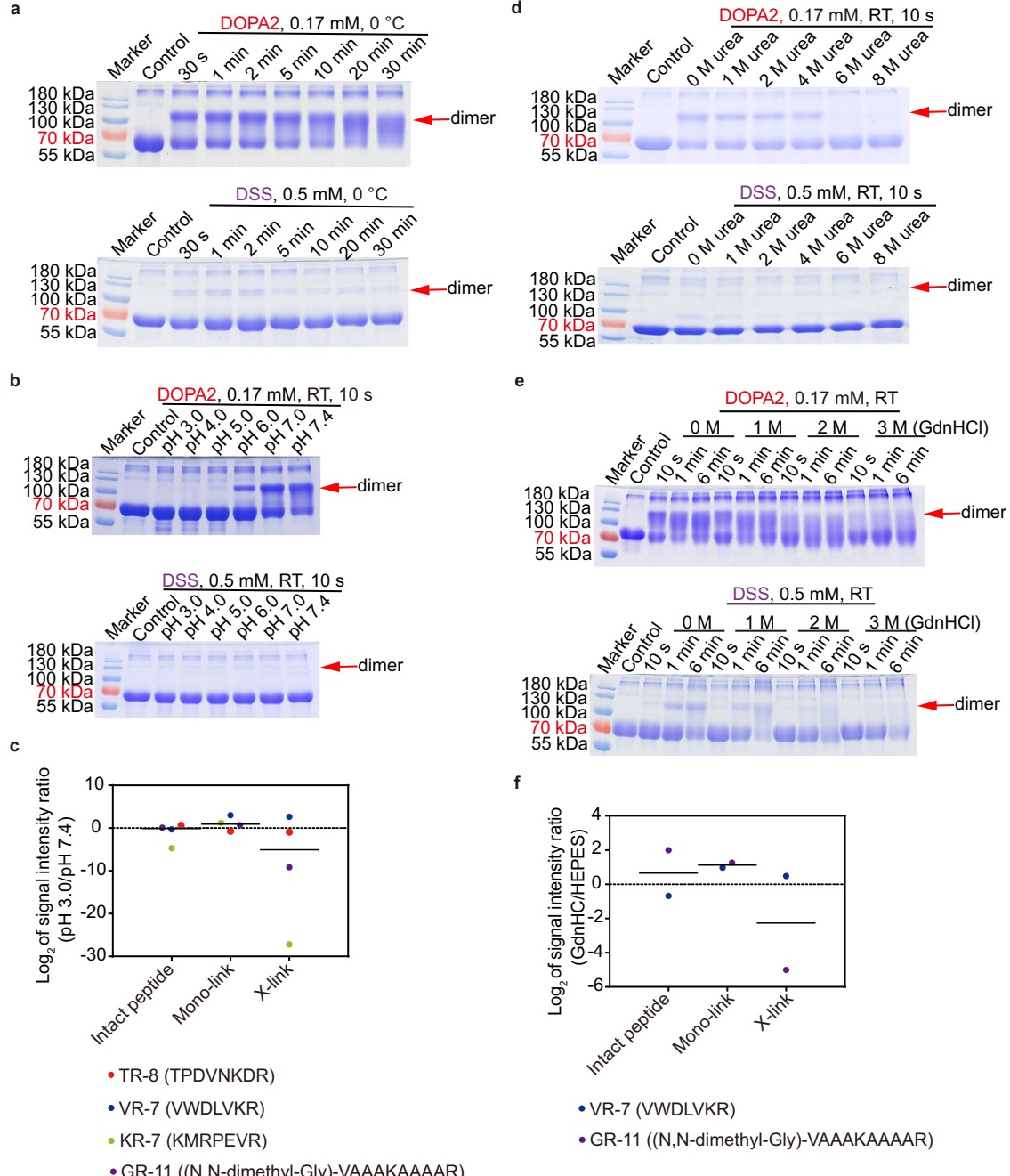

**Fig. 4 Unique properties of DOPA2 compared with DSS. a** SDS-PAGE of DOPA2 or DSS-cross-linked BSA under different reaction times at 0 °C. **b** SDS-PAGE of DOPA2 or DSS-cross-linked BSA under different pH conditions. One cross-linking experiment was performed for each protein sample in (**a**, **b**). **c** The log$_2$ transformed MS1 peak intensity ratios of cross-linked products at pH 3.0 vs. pH 7.4. The synthesized peptides TR-8, VR-7, KR-7, and GR-11 were further used to evaluate the reactivity of DOPA2 at low pH without the influence of tertiary structure. The N-terminus of GR-11 was blocked by dimethylation, and thus only the lysine residue could be cross-linked. "Intact peptide" refers to free peptides without cross-linking. "X-link" refers to the situation wherein two peptides are linked with one molecule of DOPA2. "Mono-link" refers to a peptide that has been modified but is not cross-linked by a cross-linker. $n = 4$ biologically independent samples. **d**, **e** SDS-PAGE of DOPA2 or DSS-cross-linked BSA in the presence of the indicated denaturants (urea or GdnHCl). The experiments were repeated twice independently with similar results. **f** The log$_2$ transformed MS1 peak intensity ratios of cross-linked products at GdnHCl (6 M) vs. a physiological buffer (HEPES, pH 7.4). Similar to (**c**), the synthesized peptides VR-7 and GR-11 were used to further evaluate the reactivity of DOPA2 under high concentrations of GdnHCl without the influence of tertiary structure. $n = 2$ biologically independent samples. The label of products is the same as in (**c**). Source data for a-f are provided as a Source Data file.

cross-links were hardly affected up till 4 M urea, and all except three were within-domain cross-links.

Clusters 3–5 were the opposite of Clusters 1 and 2 (Fig. 6d–f). As the concentration of urea increased to 8 M, cross-links of Cluster 3–5 either gradually or eventually but abruptly

became the dominant species. Many of them are over-length cross-links if mapped to the native structure of BSA (marked by red line, Fig. 6e, f), and the two-linked residues are typically not far apart in the primary sequence (98% <100 aa, 71% <50 aa).

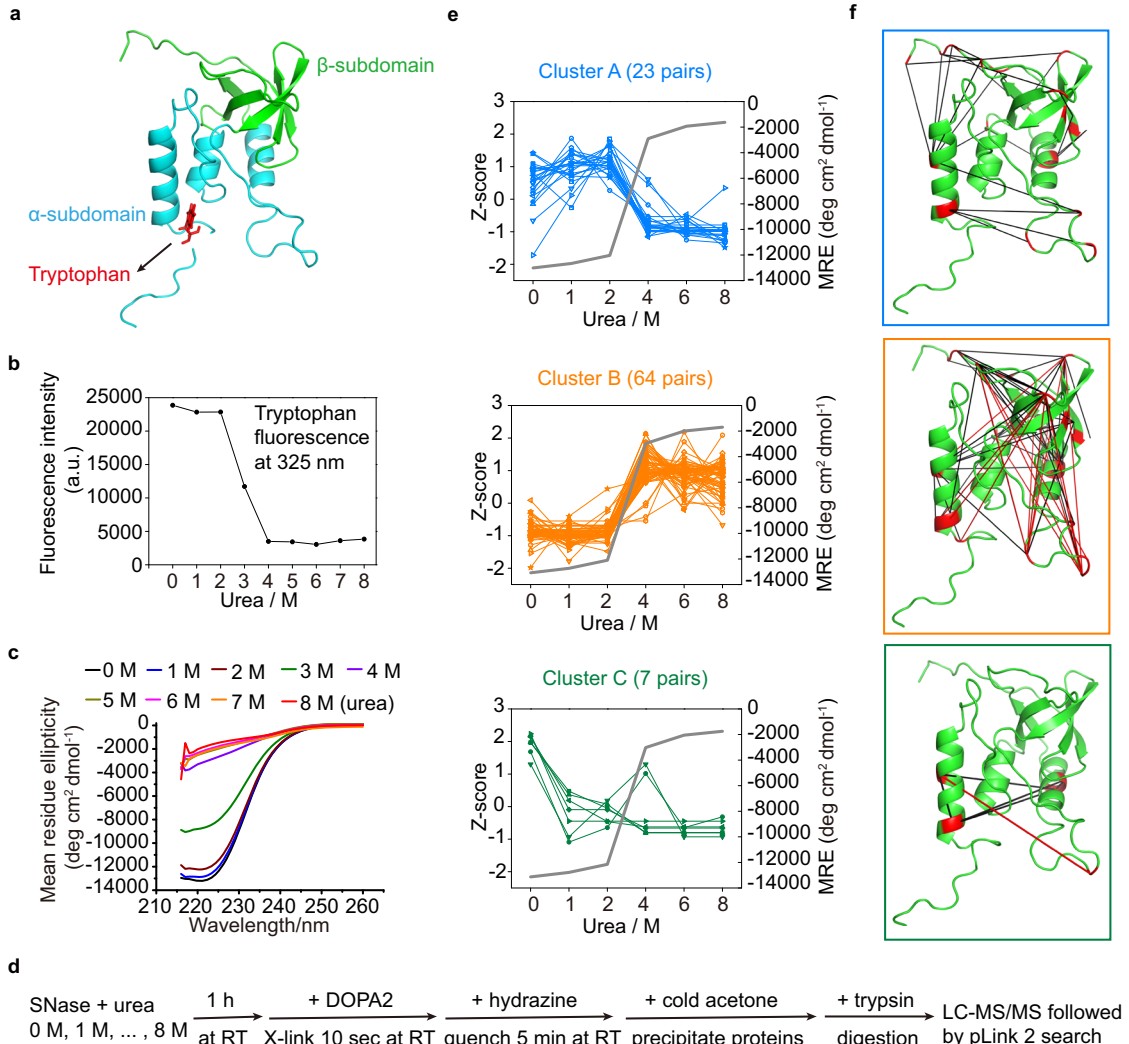

**Fig. 5 Analysis of the unfolded states of SNase by DOPA2 cross-linking. a** The crystal structure of SNase (PDB code: 1JOO[71]). **b** The unfolding transition of SNase in different concentrations of urea monitored by fluorescence emission at 325 nm. In the *y*-axis, a. u. stands for arbitrary units. **c** Circular dichroism spectra in far-UV region measured for SNase in the presence of urea. **d** Experimental workflow of cross-linking SNase in different concentrations of urea using DOPA2. **e** The changes of spectral counts for each identified cross-linked residue pair (normalized across the six conditions, shown as Z-scores on the left axis), and the mean residue ellipticity (MRE) of SNase monitored by CD at 222 nm (on the right axis) in different concentrations of urea. The residue pairs were classified into three clusters by K-means (Cluster A, Cluster B, and Cluster C). **f** The cross-links identified in different concentrations of urea were mapped on the crystal structure of RNase A (PDB code: 1JOO[71]). Cluster A is framed in blue (23 pairs), Cluster B in orange (64 pairs), and Cluster C in green (7 pairs). A red line denotes that the distance between two cross-linked residues exceeds the maximal cross-linking distance of DOPA2, and a black denotes that it does not. We performed two independent cross-linking experiments for each sample, and each was analyzed twice by LC-MS/MS. Cross-linking residue pairs were filtered by requiring FDR < 0.01 at the spectra level, *E*-value < 1 × 10$^{-8}$, and spectral counts > 3. Source data for (**b**, **c**, **e**) are provided as a Source Data file.

Of the cross-links described above, Cluster 2 and Cluster 4, each showing a sharp transition between 4 M and 6 M urea, highlight the main unfolding phase of BSA and match up the reported intermediate unfolding states of BSA. The rest of the cross-links, including four unclassified cross-links (Supplementary Fig. 9), bring details not accessed by other techniques.

**Analyzing the unfolded states of RNase A by DOPA2 cross-linking.** We also used DOPA2 to probe partially or fully unfolded states of bovine pancreatic ribonuclease A (RNase A), another classic model for protein unfolding/refolding studies[56–60]. Similar experiments on RNase A denatured in either 0–8 M urea or 0–6 M GdnHCl were conducted (Fig. 7a). To help interpret the CXMS results, we also analyzed in parallel protein conformation by CD under each of the conditions tested (Fig. 7b, c).

Cross-linking of RNase A by DOPA2 in the presence of 0–8 M urea yielded three classes of cross-links (Fig. 7d). Cluster A represents the high-in-the-native-conformation cluster (native cluster), consisting of DOPA2-linked residue pairs that slowly decrease in abundance from 0 M to 6 M urea, followed by a large drop between 6 M and 8 M urea. Cluster B, the high-in-the-non-native-conformation cluster (non-native cluster), behaves in the opposite way: the relative abundance of these DOPA2-linked lysine pairs remained low in urea concentrations below 6 M, followed by a small increase in 6 M urea and a large increase in 8 M urea. Cluster C, which has only three members, displays a U-shaped curve as the urea concentration increases.

The CXMS result and the CD result were in perfect agreement with each other. There was little change of the MRE at 222 nm

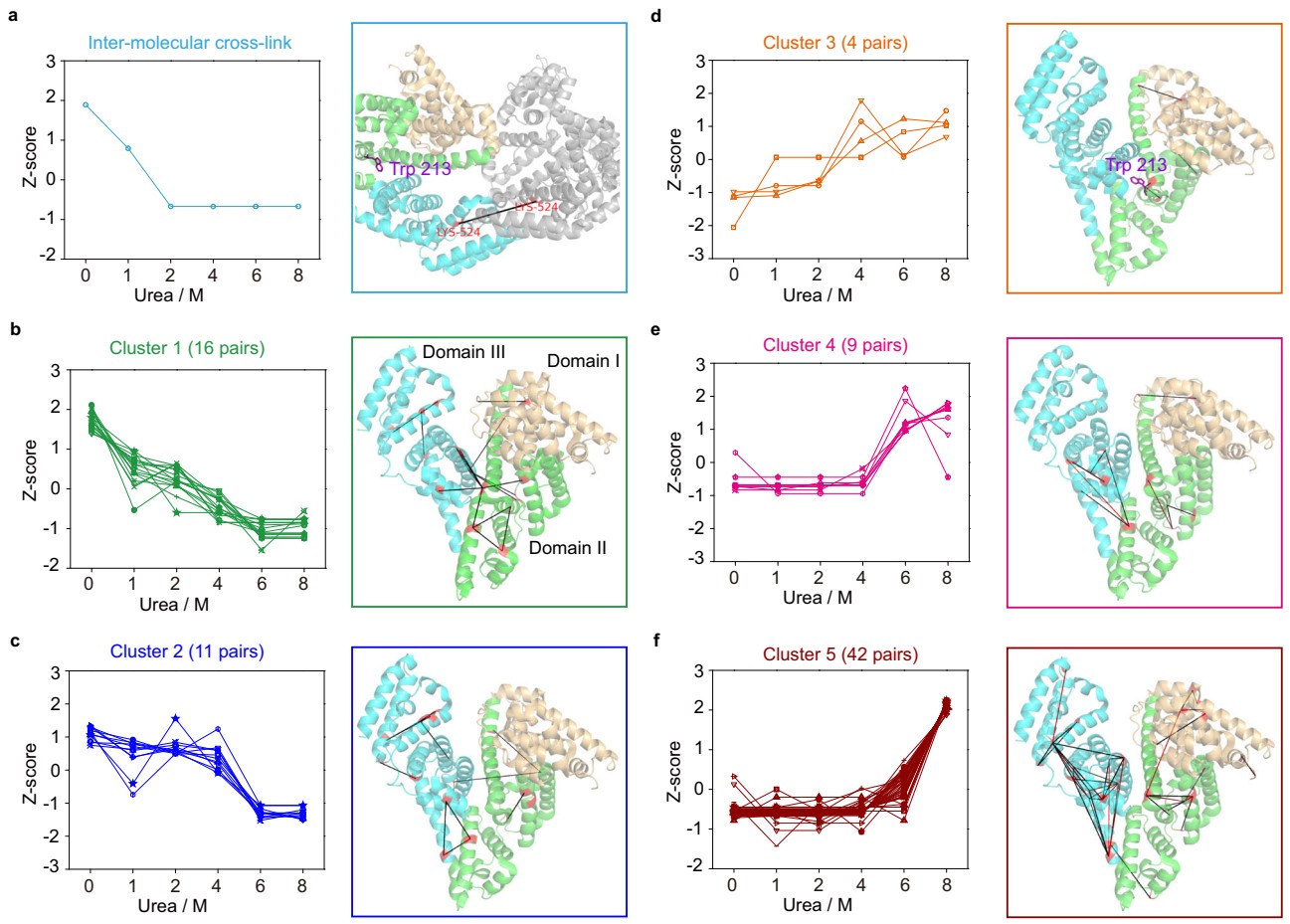

**Fig. 6 Analysis of the unfolded states of BSA by DOPA2 cross-linking. a** The spectral counts of the inter-molecular cross-link (Lys524-Lys524) are normalized as Z-scores and plotted against urea concentration. This cross-link is marked on the crystal structure of a BSA dimer (PDB code: 3V03[51]), with Trp213 represented by purple sticks in one of the subunit. **b–f** As in (**a**), but for cross-links of Clusters 1–5. The cross-links are indicated on the structure of the BSA monomer, with Domain I, II, and III in light orange, green, and cyan, respectively. The color of a two lysine residues (highlighted in red) denotes whether or not the indicated cross-link fits within the maximal cross-linking distance of DOPA2 when mapped to the native structure of BSA (black, ≤30.2 Å; red, >30.2 Å). We performed two independent cross-linking experiments for each sample, and each was analyzed twice by LC-MS/MS. Cross-linking residue pairs were filtered by requiring FDR < 0.01 at the spectra level, E-value < 1 × 10$^{-8}$ and spectral counts > 3. Source data for (**a–f**) are provided as a Source Data file.

from 0 M to 6 M urea, then an increase at 7 M and 8 M urea (Fig. 7b and y-axis values on the right in Fig. 7d), indicating a substantial loss of secondary structures between 6 M and 8 M urea. Nevertheless, RNase A in 8 M urea did not unfold completely, for its MRE values (−4749 deg cm² dmol$^{-1}$) are far from the MRE value of unfolded RNase A (−1793 deg cm² dmol$^{-1}$) in >4 M GdnHCl (Fig. 7c).

GdnHCl is a stronger denaturant than urea, and the GdnHCl concentration at 50% denaturation of RNase A is ~3 M based on the MRE at 222 nm (−4683 deg cm² dmol$^{-1}$) (Fig. 7e, gray line, y-axis values on the right). In agreement with the CD data, DOPA2-assisted CXMS analysis of RNase A in 0–6 M GdnHCl revealed that the abundance decrease of the native cluster (Fig. 7e, in blue, y-axis on the left) and the abundance increase of the non-native cluster (Fig. 7e, in orange, y-axis values on the left) quickly reached the maximum levels in 3 M GdnHCl. Strikingly, most of the cross-links maintained their abundance profiles from urea to GdnHCl. Only one cross-link belonging to the U-shaped cluster in the urea dataset had a change of profile in the GdnHCl dataset (Fig. 7e, the single magenta line in Cluster A). The above results suggest that DOPA2-assisted CXMS can be used as a general tool to study protein unfolding induced by urea or GdnHCl, despite the different properties of these two denaturants.

A closer examination of the native cluster finds that the DOPA2 cross-links in RNase A involve mainly lysine residues located on either Helix I (amino acid residues 3–13, colored by cyan) or on two surface loops (amino acid residues 34–43 and 64–72, colored by cyan) (Fig. 7f, upper panel), suggesting that these regions are most sensitive to urea. The CD data indicate that Helix I itself is largely intact in 0–4 M urea (Fig. 7b), whereas the cross-links between Helix I and either of the two loop regions decrease in abundance as the urea concentration increases from 0 M to 4 M (Fig. 7d). We thus propose that this decrease results from an increased separation between Helix I and the two surface loops, which in turn results from deformation or displacement of three loop regions in urea—the two above plus the one (amino acid residues 14–33) connecting Helix I to the rest of the protein.

Consistent with previous findings[61], we also found that the four pairs of disulfide bonds of RNase A (colored dark blue in Fig. 7f) helped to define the path of unfolding. Once these disulfide bonds were disrupted by a reducing reagent, with or without subsequent alkylation, the cross-links displayed completely different abundance profiles from the ones seen with non-reduced RNase A (compare Supplementary Fig. 10 with Fig. 7d).

In the non-native cluster, three lysine pairs Lys1-Lys98, Lys61-Lys91, and Lys1-Lys61 (Fig. 7f, middle panel) caught our attention,

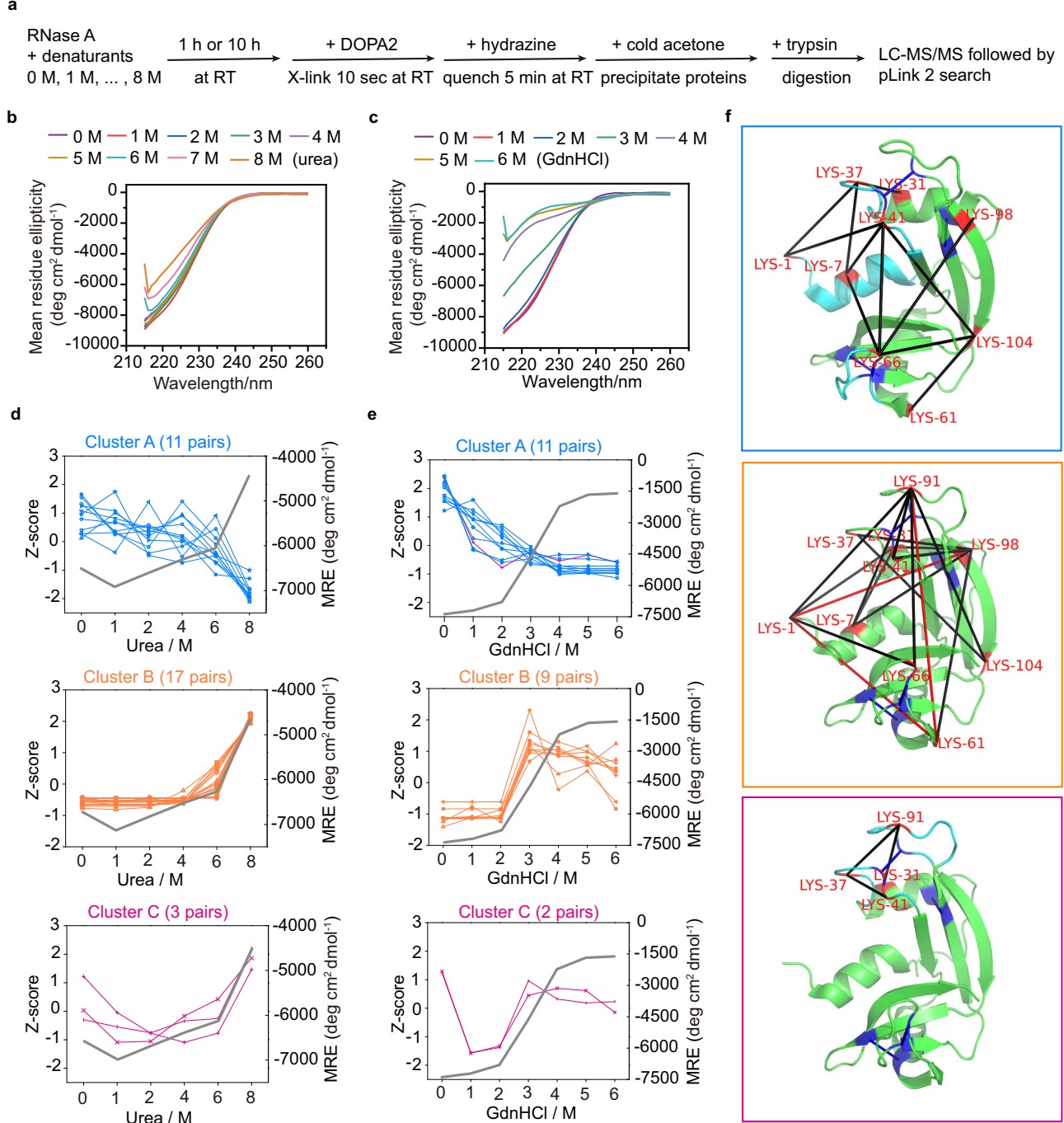

**Fig. 7 Analysis of the unfolded states of RNase A by DOPA2 cross-linking. a** Experimental workflow of cross-linking RNase A in different concentrations of denaturants (urea or GdnHCl) using DOPA2. **b**, **c** Circular dichroism spectra in far-UV region measured for RNase A in the presence of denaturants. **d** The changes of spectral counts for each identified cross-linked residue pair (normalized across the six conditions, shown as Z-scores on the left axis) and the mean residue ellipticity (MRE) of RNase A monitored by CD at 222 nm (on the right axis) in different concentrations of urea. The residue pairs were classified into three clusters by K-means (Cluster A, Cluster B, and Cluster C). **e** As in (**b**), but in different concentrations of GdnHCl. **f** The cross-links identified in different concentrations of urea were mapped on the crystal structure of RNase A (PDB code: 6ETK[68]). Cluster A in blue frame (11 pairs); Cluster B in orange frame (17 pairs); and Cluster C in magenta frame (3 pairs). The black lines denote that the distance of cross-linked residue pairs is within the restraints of cross-linkers, while the red lines denote that the distance of cross-linked residue pairs is out of the restraints of cross-linkers. The four pairs of disulfide bonds are colored dark blue. We performed two independent cross-linking experiments for each sample. Each cross-linking reaction was analyzed twice by LC-MS/MS. Cross-linking residue pairs were filtered by requiring FDR < 0.01 at the spectra level, *E*-value < 1 × 10$^{-8}$, and spectral counts > 3. Source data for (**b**–**e**) are provided as a Source Data file.

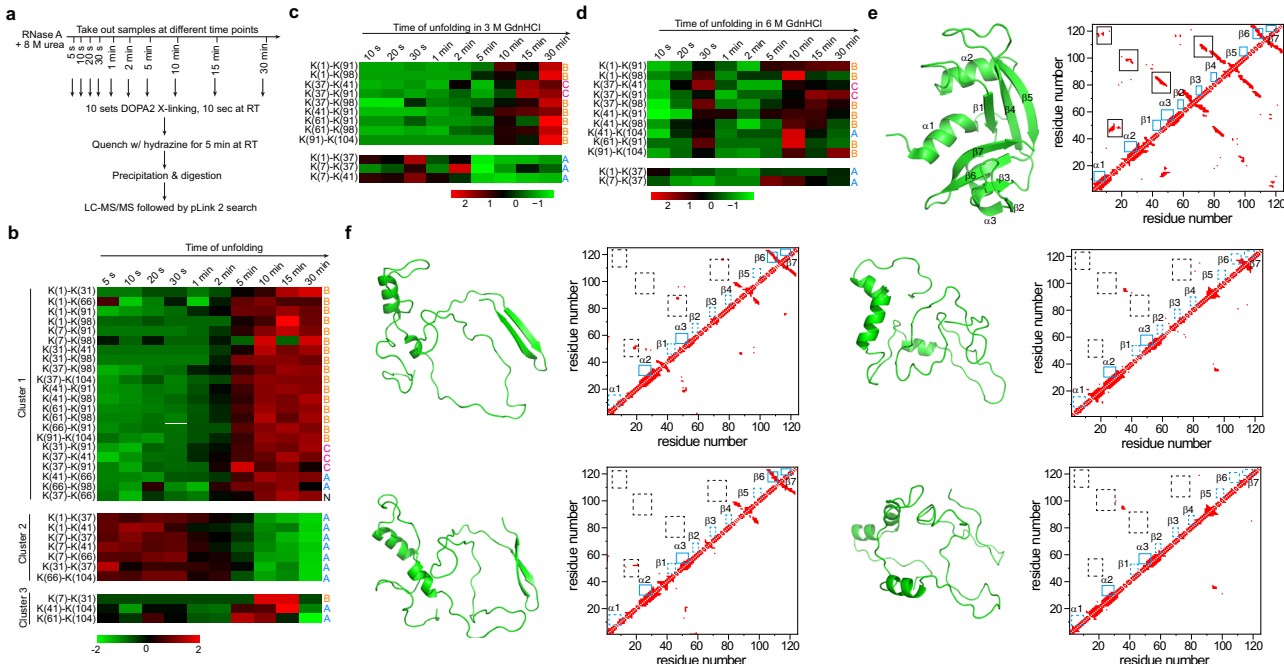

**Fig. 8 Analysis of the RNase A unfolding process by DOPA2 cross-linking. a** Experimental workflow of DOPA2 cross-linking of RNase A after it was exposed to 8 M urea for the indicated amount time. **b** Heatmap representation of the abundance changes of the identified cross-linked residue pairs. Spectral counts were used to estimate abundance changes. The cross-linked residue pairs were classified into three K-means clusters. "N" means that the cross-link did not make the cutoff for identification in the experiment of Fig. 7d. Cross-linking residue pairs were filtered by requiring FDR < 0.01 at the spectra level, $E$-value < $1 \times 10^{-8}$, and spectral counts > 3. **c** As in (**b**), but in 3 M GdnHCl. **d** As in (**b**), but in 6 M GdnHCl. Cross-linked residue pairs in (**c**, **d**) were filtered by requiring FDR < 0.01 at the spectra level, $E$-value < $1 \times 10^{-4}$, and spectral counts > 3. **e** The crystal structure of RNase A (PDB code: 6ETK[68]) and the contact map of native RNase A. The secondary structures of RNase A are labeled with blue boxes, while the mainly tertiary structures are labeled with black boxes. **f** Four categories of simulated structures of RNase A in 8 M urea. Compared with the native structure, the lost secondary and tertiary structures are represented by dashed blue boxes and black boxes, respectively. Source data for (**b–d**) are provided as a Source Data file.

because the two residues in each pair are relatively far apart in the primary sequence, as well as in 3D space in the native state. Indeed, these over-length cross-links were hardly detectable from 0 M to 4 M urea, but their numbers spiked in 8 M urea. This indicates that RNase A in 8 M urea is not completely unstructured, thus corroborating the CD result.

**Investigating the time course of RNase A unfolding by DOPA2 cross-linking**. Having established that DOPA2 cross-linking can probe the native, partially or fully unfolded states of a protein, we further studied the kinetics of protein unfolding, i.e., the time-dependence of the transition from the native state to the non-native state. As shown in Fig. 8a, RNase A was exposed to 8 M urea for varying amounts of time from 5 s to 30 min, followed by 10 s of DOPA2 cross-linking. The samples were then quenched and processed for LCMS analysis. After identifying cross-linked peptides through pLink 2 searches, we performed cluster analysis to classify the cross-links according to their abundance changes during the unfolding process (Fig. 8b). Cluster 1 consisted of 22 DOPA2-linked residue pairs whose relative abundance at 8 M urea was low in the 2 min or so and high afterwards. Cluster 2, consisting of seven members, showed an opposite profile with a sharp decrease in abundance after about 2 min of urea exposure. Cluster 3 was small and less informative. Of note, Cluster 1 is highly enriched (16/22) for members of the non-native cluster (Fig. 8b and Fig. 7d, middle panel), and Cluster 2 is made up entirely of members of the native cluster (Fig. 8b and Fig. 7d, upper panel). This shows that many cross-links behave similarly in equilibrium and kinetic unfolding of RNase A.

We observed similar kinetic profiles of RNase A unfolding in 3 M GdnHCl, albeit only a subset of the cross-links seen in the urea experiment were identified (Fig. 8c). This accords well with the CD result, for similar amounts of secondary structures remained in 8 M urea or 3 M GdnHCl as indicated by the MRE values at 222 nm (Fig. 7b, c).

When the same kinetic experiment was performed in 6 M GdnHCl, which abolished most or all secondary structures of RNase A according to CD analysis (Fig. 7c), fast and drastic changes occurred (Fig. 8d). Interestingly, a turning point appeared when RNase A was exposed to 6 M GdnHCl for 30 s: up to this point the abundance changes of the identified cross-links appeared like an accelerated version of those occurring over 30 min in 8 M urea or 3 M GdnHCl, but after this point the abundance changes reversed in direction for most of the cross-links. Also notable is that cross-links of the non-native cluster (Fig. 7d, e, middle panel) dominate the kinetic profile, and the presence of a kinetic turning point at 30 s (Fig. 8d) bears an intriguing resemblance to a steady-state turning point at 3 M GdnHCl (as shown in Fig. 8c). This is in line with the proposed biphasic unfolding model for RNase A[62].

We also performed a molecular dynamics (MD) simulation to visualize the unfolding intermediates of RNase A in 8 M urea. Snapshots were taken from the MD trajectories that can account for the non-native cross-links (Fig. 7d, middle panel), but not those of the native cross-links (Fig. 7d, upper panel). As shown in the Supplementary Movie 1 and pointed out in Fig. 8e and Supplementary Data 2, Helix 1 and three β-strands (β1, β4, and β5) disintegrate quickly upon exposure to 8 M urea. Structural elements, however, remained in dynamic equilibrium with 55.2 ± 4.5% α-helix and 27.1 ± 3.5% β-sheet on average, largely consistent with the CD results (Fig. 7b). We clustered the simulated structures of RNase A in 8 M urea into four categories

using a 3 Å cutoff (Fig. 8f). Indeed, the region around Helix 2 and Helix 3 in the native structure has a strong tendency to form helical structures even in the presence of 8 M urea (Fig. 8f).

## Discussion

In this study, we have developed an amine-selective non-hydrolyzable di-*ortho*-phthalaldehyde cross-linker, DOPA2. It offers fast cross-linking under extreme conditions (8 M urea or 6 M GdnHCl) where the commonly used NHS ester cross-linker becomes ineffective. Our results demonstrate that DOPA2-based CXMS provides a unique tool for extracting information on conformational states (including partially and fully unfolded states) as well as on continuous conformational changes associated with protein unfolding. DOPA thus opens a new application area for CXMS that could re-energize protein unfolding studies.

Many DOPA2 cross-links were identified from BSA, SNase and RNase A in the presence of 8 M urea, indicating that all three proteins possess some spatial conformation in the denatured states. SNase in particular is completely unfolded in 8 M urea according to previous studies[63], which is confirmed by the CD data at 222 nm (MRE reaching $-2000 \deg cm^2 dmol^{-1}$ at 8 M urea as shown in Fig. 5e). RNase A is not completely unfolded in 8 M urea (MRE around $-4700 \deg cm^2 dmol^{-1}$; Fig. 7b) unless its disulfide bonds are broken up by a reducing agent.

However, either with or without disulfide bonds, DOPA2 cross-links were identified from RNase A in 8 M urea (Fig. 7 and Supplementary Fig. 10). Hence, the DOPA2 cross-linking data suggest that even in the fully unfolded state, a protein is not a fully extended one-dimensional molecule at all time. In this regard, DOPA2 may also be suitable to investigate the conformational ensemble of intrinsically disordered proteins.

A closer look at the native (colored blue or green) versus non-native (denoted by warm colors) clusters of DOPA2 cross-links of BSA (Supplementary Fig. 11) finds that the non-native cross-links are concentrated within individual domains, most notably in Domain III (Supplementary Fig. 11f). For SNase and RNase A, both single-domain proteins, the lysine residues in the non-native clusters are scattered from the N- to the C-terminus (Supplementary Fig. 12).

Comparing the native and the non-native clusters in terms of the amino acid distance between two-linked residues, we find that cross-links belonging to the native clusters tend to bridge across a much longer polypeptide chain than those of the non-native clusters (Supplementary Fig. 13). For SNase and BSA, the DOPA2 cross-links in the native clusters bridge across 65 and 113 aa on average, respectively, which are 91–197% longer than the non-native clusters. RNase A is an exception: its non-native clusters have four cross-links that connect two N-terminal lysine residues (Lys1 and Lys7) with two C-terminal lysine residues (Lys91 and Lys98) (Supplementary Fig. 12b and Fig. 7f). These long-distance cross-links elevated the average amino acid distance between cross-linked residues to 51 for the non-native clusters, which is greater than the average of 37 residues for the native cluster. When all the data are combined, it is evident that the spatial contacts formed by proteins in 8 M urea are limited to local regions (90% within 75 residues); long-distance interactions (≥ 90 residues apart) occur only occasionally.

DOPA2 cross-links proteins fast (Fig. 2 and Supplementary Fig. 4c, d). We succeeded in monitoring RNase A unfolding in time intervals as short as 5 s by subjecting each time point sample to DOPA2 cross-linking for only 10 s (Fig. 8a, b). It seems that 10 s may be the minimum reaction time. As shown in Fig. 2e, when the reaction time was further reduced to 5 s, the number of cross-link spectra halved and mono-link spectra outnumbered

cross-link spectra by threefold. This is understandable because in most or all cross-linking reactions, one end of a cross-linker makes a covalent attachment before the other. Extrapolating out the trend line, it is likely that a reaction time of DOPA2 cross-linking below 1 s would generate very few cross-links.

As shown in Fig. 2h, although the OPA-amine reaction is only 3.5–7.7 times faster than the NHS ester-amine reaction, protein cross-linking by DOPA2 is 60–120 times faster than that by DSS. Possibly in a protein cross-linking reaction, which involves two consecutive OPA or NHS ester reactions, the success of the first reaction likely accelerates the second one by reducing the degree of freedom, i.e., creating a proximity effect. In other words, reactions on the two ends of a cross-linker are cooperative, with the planting step facilitating the following cross-linking step. This may account for the larger difference in the reaction speed between DOPA and DSS than that between OPA and an NHS ester.

## Methods

**Chemical instrumentation and methods**. All reactions were carried out in oven-dried glassware under an argon atmosphere, unless otherwise stated. Air and moisture sensitive reagents were transferred by syringe or cannula. Brine refers to a saturated aqueous solution of NaCl. Analytical thin layer chromatography was performed on 0.25 mm silica gel 60-F plates, and visualized using 254 nm UV light, or by staining with potassium permanganate or phosphomolybdic acid and heat as developing agents. Flash chromatography was performed using 200–400 mesh silica gels. Yields refer to chromatographically and spectroscopically pure materials, unless otherwise stated.

$^1$H NMR spectra were recorded on a Varian 400 or 500 MHz spectrometer at room temperature with CDCl$_3$ as the solvent, unless otherwise stated. $^{13}$C NMR spectra were recorded on a Varian 100 or 125 MHz spectrometer (with complete proton decoupling) at ambient temperature. Chemical shifts are reported in parts per million relative to chloroform ($^1$H, δ 7.26 ppm; $^{13}$C, δ 77.16 ppm). Data for $^1$H NMR are reported as follows: chemical shift, integration, multiplicity (s = singlet, d = doublet, t = triplet, q = quartet, m = multiplet) and coupling constants. High-resolution mass spectra were obtained at Peking University Mass Spectrometry Laboratory using a Bruker APEX instrument.

Synthetic compounds were analyzed by UPLC/MS on a Waters UPLC H Class and SQ Detector 2 system. The system was equipped with a Waters C18 1.7 μm Acquity UPLC BEH column (2.1 × 50 mm), equilibrated with HPLC grade water (solvent A) and HPLC grade acetonitrile (solvent B) with a flow rate of 0.3 mL/min.

**Reagents and solvents**. All chemical reagents were from J&K, Alfa Aesar, and TCI Chemicals without further purification unless otherwise stated. DCM and CH$_3$CN were distilled from calcium hydride. THF was distilled from sodium/benzophenone ketyl.

For the MS analysis, cross-linkers (DSS, EDC, SDA, and sulfo-LC-SDA), tris(2-carboxyethyl) phosphine (TCEP), Sulfo-NHS, and 2-Iodoacetamide (IAA) were purchased from Pierce Biotechnology (Thermo Scientific). Cross-linker Bis- PEG1-NHS ester (BSMEG) was purchased from BroadPharm. Guanidine hydrochloride (GdnHCl) was purchased from MP Biomedicals LLC. Dimethylsulfoxide (DMSO), HEPES, NaCl, KCl, urea, CaCl$_2$, methylamine, and other general chemicals were purchased from Sigma-Aldrich. Acetonitrile (ACN), formic acid (FA), acetone, and ammonium bicarbonate were purchased from J.T. Baker. Trypsin and Asp-N (gold mass spectrometry grade) were purchased from Promega.

**Synthesis of DOPA**. The synthetic procedures and NMR spectra of DOPA-C$_2$, DOPA1, and DOPA2 are provided in Supplementary Methods, Supplementary Figs. 15–26.

**Preparation of protein samples**. Aldolase, BSA, catalase, lysozyme, myosin, lactoferrin, carbonic anhydrase 2, and β-amylase were obtained from Sigma-Aldrich. RNase A was purchased from Thermo Fisher. Recombinant GST containing an N-terminal His tag was expressed in *E. coli* BL21 cells from the pDYH24 plasmid and purified with glutathione sepharose (GE Healthcare). PUD-1/2 heterodimers were purified on a HisTrap column followed by gel filtration. Stock solutions of model proteins were individually buffer exchanged into 20 mM HEPES, pH 8.0 by ultrafiltration. The mixture of ten proteins (aldolase, BSA, catalase, carbonic anhydrase 2, lysozyme, lactoferrin, β-amylase, myosin, GST, and PUD-1/2) was prepared at a final concentration of 0.1 mg/mL (total protein concentration, 1 mg/mL).

The DNA sequence of cameleon calcium sensor protein (YC3.6) was cloned into pETDuet-1 vector. The N-terminal of YC3.6 was fused with a 6×His tag for affinity purification. The His$_6$-tagged YC3.6 was expressed in *E. coli* BL21 cells. Cells were lysed using an Ultrasonic Cell Disruptor in lysis buffer (50 mM HEPES,

pH 7.4, 0.3 M NaCl, 10 mM imidazole, 10% glycerol, 1 mM PMSF), and purification was performed using Ni-NTA-agarose (QIAGEN). The eluate was buffer exchanged into storage buffer (20 mM HEPES, pH 7.4, 0.15 M NaCl, 10% glycerol) before freezing at −80 °C. The purity of the YC3.6 protein was analyzed on SDS-PAGE which revealed a single band of ~75 kDa without degradation.

Recombinant SNase (V8 type) was purified as previously described[48]. Briefly, *E. coli* cells with induced expression of SNase, which formed inclusion bodies, were sonicated in lysis buffer (50 mM Tris-HCl, pH 9.2, 2 mM EDTA, 2% Triton X-100, 0.5 mM PMSF) and centrifuged at 22,000 × *g* for 15 min at 4 °C. The supernatant was discarded and the pellet was washed carefully to remove the membrane-rich top layer. The washed pellet, which contained the inclusion bodies, was resuspended in lysis buffer and centrifuged again at 22,000 × *g* at 4 °C for 15 min. After two more washes, the pellet was redissolved in 6 M urea, 50 mM Tris-HCl, pH 9.2. The solution was cleared by centrifugation and loaded onto a CM-25 carboxymethyl-Sephadex column. Proteins were eluted with 6 M urea, 0.5 M NaCl, 50 mM Tris-HCl, pH 9.2. The fractions making up the main peak of SNase was pooled, dialyzed extensively against water, lyophilized, and redissolved in 20 mM HEPES, 150 mM NaCl, pH 7.4 before further purification using a Superdex 75 size exclusive column. The purity of the SNase sample thus prepared was estimated to be >95% by SDS-PAGE.

**Kinetics experiments for second-order rate constant determination**. Equation for calculating the second-order rate constants is shown below.

$$\ln \frac{(a-x)}{(b-x)} = k(a-b)t + \ln \frac{a}{b} \qquad (1)$$

where a represents the amount of OPA/NHS ester analog in the beginning, b represents the amount of Boc-OMe-lysine in the beginning, *x* represents the consumption of OPA/NHS ester analog/Boc-OMe-lysine at time *t* (the consumption is equivalent), (a−x) represents the amount of remaining OPA/ NHS ester analog at time *t* (this amount is equal to the amount of quenched product), (b−x) represents the amount of remaining Boc-OMe-lysine at time *t*, and k represents the second-order reaction kinetic constants.

*Kinetics experiments of Boc-OMe-lysine with OPA*. Boc-OMe-lysine (5 μmol, 1 eq.) was dissolved in 10 mL TEAB buffer at 25 °C. OPA (10 μmol, 2 eq.) was added to the stirred reaction mixture. The reaction was monitored at 0.5 min, 1 min, 1.5 min, 2 min, 2.5 min, 3 min, 3.5 min, 4 min. At each time point, a 10 μL reaction mixture was quenched with 1 μL 98% hydrazine. The conversion was monitored by LC-MS.

*Kinetics experiments of Boc-OMe-lysine with NHS ester analog*. Boc-OMe-lysine (5 μmol, 1 eq.) was dissolved in 10 mL TEAB buffer at 25 °C. NHS (10 μmol, 2 eq.) was added to the stirred reaction mixture. The reaction was monitored at 2 min, 3 min, 4 min, 5 min, 6 min, 7 min, 8 min, 9 min. At each time point, a 10 μL reaction mixture was quenched with 1 μL 50% hydroxylamine. The conversion was monitored by LC-MS.

**FRET-based comparison of the reaction rate of Cy5-OPA and Cy5-NHS ester with peptide Cy3-K**. The N- and C-termini of fluorescently labeled peptide Cy3-K (sequence: Cy3-CGGAAGKVGR) were blocked with acetylation and amidation, respectively. The only reactive group was the ε-NH$_2$ of the lysine residue. A dilution series of both Cy5-OPA and Cy5-NHS ester (5.000, 2.500, 1.250, 0.625, 0.313, 0.156, and 0.078 μM) were prepared for fluorescence measurements in a 96 well plate using an EnSpire Multimode Plate Reader to generate standard curves of fluorescence intensity vs. Cy5-OPA and Cy5-NHS ester concentration (excitation/ emission = 649/666 nm). For the conjugation of Cy3-K and Cy5-OPA or Cy5-NHS ester, Cy3-K was dissolved in 20 mM HEPES, 150 mM NaCl, pH 7.4 at a final concentration of 25 μM. Cy5-OPA or Cy5-NHS ester was added to a final concentration of 2.5 μM. The reactions were monitored continuously for 30 min with an interval of 20 s (excitation/emission = 525/666 nm). The concentration of conjugated Cy3-K and Cy5-OPA or Cy5-NHS ester was estimated by fitting to the standard curve established above. To estimate the reaction rate constant, we assumed that the product of the reaction (a peptide with both Cy3 and Cy5 fluorophore) has 100% energy transfer efficiency from Cy3 to Cy5 upon Cy3 excitation. FRET ratio = $(F^{\text{Reaction}} - F_0^{\text{Cy3-K}})/F_0^{\text{Cy5-OPA/NHS ester}}$, where $F^{\text{Reaction}}$ is the fluorescence intensity measured at 666 nm; $F_0$, also measured at 666 nm, is the initial fluorescence intensity of a substrate before the reaction ($t = 0$).

**FRET-based comparison of the reaction rate of DOPA2 and DSS with protein YC3.6**. Protein YC3.6 was diluted to 1 mg/mL (~13.3 μM) with HEPES buffer (20 mM HEPES and 150 mM NaCl, pH 7.4). DOPA2 and DSS were added to a final concentration of 0.17 mM and 0.5 mM, respectively. The cross-linking reaction was monitored at 5 s, 10 s, 20 s, 30 s, 40 s, 1 min, 2 min, 3 min, 5 min, 10 min, 20 min, and 1 h. At each time point, the reaction was quenched and diluted with quenching buffer (10 mM hydrazine, 50 μM EGTA, 20 mM HEPES, 150 mM NaCl, pH 7.4). The final concentration of YC3.6 for fluorescence detection was 500 nM. With an excitation of 420 nm, fluorescent emission spectra were scanned from 450 to 600 nm using an EnSpire Multimode Plate Reader or Spark Multimode

Microplate Reader. The ratio 527/480 nm was defined as the emission fluorescence intensity at 527 nm divided by the emission fluorescence intensity at 480 nm.

**Characterization of OPA selectivity towards different amino acids**. Ten synthesized peptides (listed in Supplementary Table 1) were dissolved in 20 mM HEPES, pH 7.4 to a final concentration of 2 mM. OPA was added to a final concentration of 2 mM. After a 1-h reaction at room temperature, the reaction products were analyzed by LC-MS/MS.

**Peptide cross-linking**. The synthesized peptides VR-7 (sequence: VWDLVKR), KR-7 (sequence: KMRPEVR), TR-8 (sequence: TPDVNKDR), and GR-11 (sequence: (N,N-dimethyl-Gly)-VAAAKAAAAR) were separately dissolved in 20 mM HEPES, pH 7.4 at a final concentration of 2 mM. Cross-linker DOPA2 was added to a final concentration of 2 mM. For testing the activity of DOPA2 under acidic pH or in the presence of high concentrations of denaturants, peptides VR-7, KR-7, TR-8, GR-11 were dissolved in 100 mM citric acid-Na$_2$HPO$_4$, pH3.0, or in 6 M GdnHCl to a final concentration of 2 mM. Cross-linker DOPA2 was added to a final concentration of 2 mM. After cross-linking at room temperature for 1 h, each reaction mixture was diluted with 0.1% FA and about 5 pmol of total peptides were analyzed by liquid chromatography coupled with tandem mass spectrometry (LC-MS/MS).

**Protein cross-linking**. The cross-linking conditions, reaction time, reaction temperature, reaction buffers, and concentrations of cross-linkers were optimized using 1 mg/mL BSA. Six model proteins were each diluted to 1 mg/mL in HEPES buffer (20 mM HEPES and 150 mM NaCl, pH 7.4). The ten-protein mixture was diluted to 1 mg/mL (total protein) in the same HEPES buffer. Each protein solution was cross-linked with DOPA-C$_2$, DOPA1, or DOPA2 (protein-to-cross-linker mass ratio at 16:1) for 10 min, or cross-linked with DSS (0.5 mM) for 1 h. For a broader comparison between DOPA2 and other cross-linkers, BSA and the ten-protein mixture were diluted to 1 mg/mL and cross-linked with BSMEG (0.5 mM) for 1 h, or cross-linked with EDC (2 mM and coupled with 5 mM Sulfo-NHS) for 2 h. Cross-linking of BSA and ten-protein mixture with SDA or Sulfo-LC-SDA was carried out in two steps. First, the protein was incubated with 1.5 mM SDA or Sulfo-LC-SDA in the dark for 1 h at room temperature. Second, the samples were spread onto the inside of Eppendorf tube lids to form a thin film, placed on ice at a distance of 5 cm from a UV lamp, and irradiated by 365 nm light for 30 min at 200,000 μJ/cm$^2$.

The reactivity of cross-linkers DOPA2 and DSS in different pH conditions was evaluated with BSA. First, BSA was dissolved in citric acid-Na$_2$HPO$_4$ buffers of pH 3.0, 4.0, 5.0, 6.0, 7.0, and 7.4 at a final concentration of 1 mg/mL, respectively. Each was then cross-linked with DOPA2 (0.17 mM) or DSS (0.5 mM) for 10 s.

Similarly, the reactivity of cross-linkers DOPA2 and DSS in different concentrations of denaturants were also evaluated with BSA. Here, BSA was dissolved to 1 mg/mL in 0, 1, 2, 4, 6, and 8 M urea at room temperature for 1 h. Each was then cross-linked with DOPA2 (0.17 mM) or DSS (0.5 mM) for 10 s. Likewise, BSA was dissolved to 1 mg/mL in 0, 1, 2, and 3 M GdnHCl at room temperature for 1 h. Each was then cross-linked with DOPA2 (0.17 mM) or DSS (0.5 mM) at room temperature for 10 s, 1 min, 6 min, respectively. All the reactions were quenched with 20 mM hydrazine at room temperature for 5 min. The cross-linking reactions of RNase A in different concentrations of urea (0, 1, 2, 4, 6, and 8 M) or in GdnHCl (0, 1, 2, 3, 4, 5, and 6 M) were similar to that of BSA. RNase A was incubated in each urea solution for 1 h and in each GdnHCl solution for 10 h before cross-linking. Reactions were timed and quenched as described above. The samples were diluted with 1× PBS buffer to a final denaturant concentration of 1.5 M (except for 0 and 1 M urea or GdnHCl). Cross-linked proteins were precipitated with four volumes of cool acetone. The cross-linking reaction of SNase in different concentrations of urea (0, 1, 2, 4, 6, and 8 M) is similar to that described in RNase A.

For the cross-linking reaction of RNase A in 8 M urea at different time points, 20 μL of 10 mg/mL RNase A was quickly mixed with 160 μL of 10 M urea and 20 μL of 1× PBS buffer, pH 7.4. Reaction samples (each 20 μL) were drawn at time points (at 5, 10, 20, and at 1, 2, 5, 10, 15, and 30 min) and terminated by the addition of 0.4 μL 1 M hydrazine at room temperature for 5 min. (Note: the first four time points were quite dense and difficult to collect continuously. Thus, these samples were prepared individually before the addition of cross-linkers.) The cross-linking reaction of RNase A in 3 M GdnHCl or 6 M GdnHCl at different time points is similar to that described in urea.

**Trypsin digestion**. Cross-linked proteins were precipitated by four volumes of acetone for at least 30 min at −20 ˚C. The pellets were air dried and then dissolved, assisted by sonication, in 8 M urea, 20 mM methylamine, 100 mM Tris, pH 8.5. After reduction (5 mM TCEP, RT, 20 min) and alkylation (10 mM iodoacetamide, RT, 15 min in the dark), the samples were diluted to 2 M urea with 100 mM Tris, pH 8.5. Denatured proteins were digested by trypsin at a 1/50 (w/w) enzyme/ substrate ratio at 37 °C for 16–18 h, and the reactions were quenched with 5% formic acid (final conc.). Wear protective gloves as methylamine can cause skin burns and eye damage.

**LCMS analysis**. All synthesized peptide samples were analyzed using an EASY-nLC 1000 system (Thermo Fisher Scientific) interfaced with a Q-Exactive HF mass spectrometer (Thermo Fisher Scientific). Peptides were loaded on a pre-column (75 µm ID, 4 cm long, packed with ODS-AQ 120 Å–10 µm beads) and separated on an analytical column (75 µm ID, 12 cm long, packed with Luna C18 1.9 µm 100 Å resin). Slight modifications to the separation method were made for different samples. The ten OPA modified peptides were injected and separated with a 30 min linear gradient at a flow rate of 200 nL/min as follows: 0–5% B in 2 min, 5–30% B in 15 min, 30–100% B in 3 min, and 100% B for 10 min (A = 0.1% FA, B = 100% ACN, 0.1% FA). The DOPA2 cross-linked peptides VR-7, KR-7, TR-8, and GR-11 were injected and separated with a 30 min linear gradient at a flow rate of 200 nL/min as follows: 0–5% B in 2 min, 5–28% B in 15 min, 28–100% B in 3 min, and 100% B for 10 min (A = 0.1% FA, B = 100% ACN, 0.1% FA). Spectra were acquired in the data-dependent mode: the top fifteen most intense precursor ions from each full scan (resolution 60,000) were isolated for HCD MS2 (resolution 15,000, NCE 27) with a dynamic exclusion time of 30 s. Precursors with more than 6+, or unassigned charge states were excluded. In order to acquire more high-quality spectra at the peaks, ten OPA modified peptides were also analyzed with a dynamic exclusion time of 10 s.

All proteolytic digestions of proteins were analyzed with the equipment listed above for synthesized peptides. Peptides were loaded on a pre-column and separated on an analytical column as noted above. Slight modifications to the separation method were made for different samples. The BSA, SNase and RNase A samples were injected and separated with a 60 min linear gradient at a flow rate of 200 or 300 nL/min as follows: 0–5% B in 2 min, 5–30% B in 43 min, 30–100% B in 5 min, and 100% B for 10 min (A = 0.1% FA, B = 100% ACN, 0.1% FA). A ten-protein mixture was injected and separated with a 90 min linear gradient at a flow rate of 300 nL/min as follows: 0–5% B in 1 min, 5–30% B in 69 min, and 30–100% B in 10 min, 100% B for 10 min (A = 0.1% FA, B = 100% ACN, 0.1% FA). The top fifteen most intense precursor ions from each full scan (resolution 60,000) were isolated for HCD MS2 (resolution 15,000; NCE 27) with a dynamic exclusion time of 30 s. Precursors with 1+, 2+, more than 6+, or unassigned charge states were excluded.

**Identification of cross-links with pLink 2**. The search parameters used for pLink 2 were as follows: instrument, HCD; precursor mass tolerance, 20 ppm; fragment mass tolerance 20 ppm; cross-linker DOPA-C$_2$ (cross-linking sites K and protein N-terminus, cross-link mass-shift 258.068, mono-link w/o hydrazine mass-shift 276.079, mono-link w/t hydrazine mass-shift 272.095); cross-linker DOPA1 (cross-linking sites K and protein N-terminus, cross-link mass-shift 290.058, mono-link w/o hydrazine mass-shift 308.068, mono-link w/t hydrazine mass-shift 304.085); cross-linker DOPA2 (cross-linking sites K and protein N-terminus, cross-link mass-shift 334.084, mono-link w/o hydrazine mass-shift 352.096, mono-link w/t hydrazine mass-shift 348.111); cross-linker DSS (cross-linking sites K and protein N-terminus, cross-link mass-shift 138.068, mono-link mass-shift 156.079); cross-linker BSMEG (cross-linking sites K and protein N-terminus, cross-link mass-shift 126.032, mono-link mass-shift 144.042); cross-linker EDC (cross-linking sites K or protein N terminus with D or E, cross-link mass-shift −18.011); cross-linker SDA (cross-linking sites K or protein N terminus with any amino acid, cross-link mass-shift 82.042); cross-linker Sulfo-LC-SDA (cross-linking sites K or protein N terminus with any amino acid, cross-link mass-shift 195.126); fixed modification C 57.021; peptide length, minimum 6 amino acids and maximum 60 amino acids per chain; peptide mass, minimum 600 and maximum 6000 Da per chain; enzyme, trypsin or trypsin and Asp-N, with up to three missed cleavage sites per cross-link. Protein sequences of model proteins were used for database searching. The results were filtered by requiring a spectral false identification rate <0.01. MS2 spectra were annotated using pLabel[64], requiring mass deviation ≤ 20 ppm.

**Cα-Cα distance calculations**. The Cα-Cα Euclidean distances were calculated by an in-house Perl script with coordinates from the PDB files. The Cα-Cα Solvent Accessible Surface Distance (SASD) was calculated using Jwalk[65] or TopoLink[66]. For model proteins, it is not possible to distinguish from the sequence of the peptide whether the cross-links are intra- or inter-molecular. We thus calculated all the possible combinations and picked those with the shortest Cα-Cα distance. If the SASD of cross-linked residue pairs could not be calculated (due to a lack of surface accessibility), we excluded them from the calculation. When calculating structural compatibility, the distance cut-offs were 24.9 Å for DOPA-C$_2$, 27.7 Å for DOPA1, 30.2 Å for DOPA2, and 24.0 Å for DSS.

**K-means clustering and data normalization**. The spectral counts of each cross-linked site pair across all samples were first transformed to Z-Score (mean centered and scaled to the variance). K-means clustering was performed using Cluster 3.0 with the settings of 3 clusters and a maximum of 100 iterations.

**Circular dichroism (CD) and fluorescence measurements**. CD spectra were measured on a Chirascan-plus circular dichroism spectrometer. Measurements were taken at 20 °C and pH 7.4. Data were recorded between 210 and 260 nm in a quartz cuvette with 1 mm path length. Measurements were acquired in 1 nm increments with an integration time of 0.5 s. RNase A was measured at 7 µM

concentration and in the presence of different concentrations of urea (0–8 M) and GdnHCl (0–6 M). SNase was measured at 70 µM concentration and in the presence of different concentrations of urea (0–8 M). Three scans were averaged for each measurement. The averaged CD spectra for one condition and its respective buffer were subtracted and then smoothed (window size = 2). The resulting CD spectra were plotted with OriginPro 8.

The intrinsic tryptophan fluorescence emission intensity of SNase was recorded on an EnSpire Multimode Plate Reader at 25 °C for three repeats. SNase was measured at 5 µM concentration and in the presence of different concentrations of urea (0–8 M). Samples were excited at 295 nm and the emission was recorded at 325 nm.

**MD simulations**. All molecular dynamics (MD) simulations were performed using AMBER20 software package[67]. We first analyzed the bond, angle as well as dihedral parameters of DOPA2 using Gaussian 09 (Gaussian, Inc.). The B3LYP/6-31 G+(d) scheme was used for the calculations. We first performed the MD simulations on DOPA2. The DOPA2 was placed in a TIP3P period water box with 10 Å padding in all directions. The simulations were run for three independent trajectories with 100 ns. The simulations were performed at 298 K with non-bond interactions cutoff of 10 Å. The distance distribution of DOPA2 was calculated using CPPTRAJ module in AMBER20. The simulation results show that the maximum arm length of DOPA2 is about 30 Å (for Cα-Cα atoms) (Supplementary Fig. 14)

We performed explicit solvation MD simulations on RNase A with distance restraints from cross-links using the same way described above. The corresponding Cα-Cα distance (involved in cross-linked residues) will be calculated during the simulation to construct the energy function. The crystal structure, PDB code 6ETK[68], was used as the starting conformation for the simulations. The simulations were performed at 298 K for 500 ns. Three clusters of cross-links were identified during the unfolding process of RNase A under 8 M urea. It can be seen from the experimental results that the number of spectra in Cluster A gradually decreases with the increase of urea concentration, while Cluster B is just the opposite. Therefore, in our simulation process, the strategies for setting the distance restraints were: the distance corresponding to the residues in Cluster A should be larger than the arm length DOPA2 (>30 Å), while the residues in Cluster B should be less than the arm length of DOPA2 (<30 Å). Consequently, the square bottom of restraints energy function is 30–100 Å for residues in Cluster A, and 4 to 30 Å for residues in Cluster B, respectively. An energy penalty will be performed when the corresponding Cα-Cα distance is larger than 100 Å (very long distance) or less than 4 Å (too close) using a parabolic function. The CPPTRAJ module was used for the analysis of trajectory. We first pick out the structures that satisfy all cross-linking restraints (>30 Å in Cluster A and <30 Å in Cluster B) from the trajectory. After that, the structures were clustered using density-based clustering algorithm[69], the minimum number of points required to form a cluster is 10, and the distance cutoff between points for forming a cluster is 3 Å. The native contact, as well as secondary structure, was also calculated using CPPTRAJ with default parameters.

**Reporting summary**. Further information on research design is available in the Nature Research Reporting Summary linked to this article.

## Data availability
The mass spectrometry raw data of DOPA CXMS analysis for BSA, SNase, and RNase A in this study were deposited to the ProteomeXchange Consortium via the iProX partner repository[70] with the dataset identifier PXD030552. All other data are available from the corresponding authors on reasonable request. Source data are provided with this paper. Protein structures used in this study have been published before and are available in the Protein Data Bank under the following accession codes 3V03, 1JOO, 6ETK, 4JDE, 1ZAH, 5GKN, 1Y6E, 1LYZ. Source data are provided with this paper.

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

## Acknowledgements

The authors gratefully acknowledge financial support from the National Key Research and Development Program of China (2017YFA0505200 X.L. and 2018YFA0507700 to C.T.), the Ministry of Science and Technology of China (SQ2020YFF010019 to M.-Q.D.), the National Natural Science Foundation of China Grant (21625201, 21961142010, and 91853202 to X.L.; 22161132013 to C.T.), the Beijing Outstanding Young Scientist Program (BJJWZYJH01201910001001 X.L.), the municipal government of Beijing (in the form of NIBS intramural grants), TIMBR, and Tsinghua University, the Program for Donglu Scholar in the Yunnan University.

## Author contributions

J.-H.W. purified proteins, performed MS, CD, fluorescence, FRET experiments, analyzed and interpreted the data, and wrote the manuscript; Y.-L.T. designed and synthesized the compounds, analyzed the data, and wrote the manuscript; Z.G. performed MD simulation experiments; M.-Q.D. guided the study, interpreted the data, and wrote the manuscript; X.L., conceived the DOPA cross-linker, guided the study, and revised the manuscript; R.J. did CD analysis of RNase A, helped with data analysis and interpretation, revised the manuscript; F.X. performed the kinetics experiments; Z.G. and C.T. helped with data analysis and interpretation; D.T. performed MS experiments; Q.L. performed the chemical synthesis; S.-Q.L. helped with data analysis and interpretation, revised the manuscript; K.Y. helped with data analysis and interpretation.

## Competing interests

The authors declare no competing interests.
