## [Peer Review File · Nature Communications]

Reviewers' comments:

Reviewer #1 (Remarks to the Author):

A Non-hydrolyzable Lysine Cross-linker DOPA Enables Mass Spectrometry Analyses of Protein Unfolding and Weak Protein-protein Interactions

Jian-Hua Wang, Yu-Liang Tang, Rohit Jain, Fan Xiao, Zhou Gong, Yu Zhou, Dan Tan,

Qiang Li, Xu Dong, C. Robert Matthews, Shu-Qun Liu, Chun Tang, Niu Huang, Keqiong Ye,

Meng-Qiu Dong, Xiaoguang Lei

In this manuscript, the authors introduce a novel crosslinking reagent, di-ortho-phthalaldehyde crosslinker, DOPA2, which was shown to be non-hydrolyzable and amine-selective. In different experimental setups this reagent proved to have 6-10 times higher reaction rates than conventional NHS- esters like DSS, tolerated high concentrations of denaturing substances like Urea and guanidine hydrochloride and thus, was found to be predestined for monitoring the folding/unfolding process of proteins in short time scales and to detect weak and specific interactions.

Overall, I appreciate the development of new crosslinking reagents as they open up new avenues for different applications in the crosslinking field. However, in my opinion the added value of this reagent was insufficiently demonstrated and thus, I do not think that this work is suitable for publication in Nature Communications. I recommend to transfer this manuscript to a more specialized journal.

Major and Minor criticism:

1. I was irritated by the different numbers of how much faster the reaction rate of DOPA was in comparison to DSS: 6x, 10x, 60x?
2. As mentioned in the discussion the DOPA reagent is sensitive to overcrosslinking in terms of reagent concentration and reaction time – this needed to be more carefully shown if you introduce a new reagent. What implications does overcrosslinking have?
3. In my opinion, the claim “monitoring the folding/refolding process of a protein” is overstating the capability of the method. As demonstrated it is able to identify residues or patches which are more or less sensitive to chaotropic agents, however, this does not necessarily describe the folding process.

4. As I said before, the demonstration of the added value, of the new insights you can achieve with this reagents is not sufficient for Nature Communications.

Reviewer #2 (Remarks to the Author):

This manuscript reports the use of di-ortho-phthalaldehyde (DOPA) as a cross-linker (XLer) for Lys and Ser residues as a replacement for NHS esters. The new cross-linking chemistry is well described and tested, including the effect of time, concentration, temperature, and the presence of chaotropic agents. The new XLer is tested against BSA, RNase A proteins and EIN/HPr and EIIA/EIIB complexes, besides a ten protein commercial mix. This is indeed an interesting contribution to the field of structural mass spectrometry.

My main issue with this manuscript is related to the relevance for the wider audience of Nature Communications when the following points:

- 1) Even though the data presented here are very interesting, it is an incremental improvement, i. e., the DOPA XLer behaves better than NHS esters but the gain are sometimes marginal or complementary in nature. I believe is a good contribution to the cross-linking field, but NHS esters can do most of what is presented here, just with a lower efficiency;
- 2) One of the main attractiveness of DOPA is the higher structural compatibility of identified cross-links compared to NHS esters (II. Performance of DOPA vs DSS on model proteins) but data is not convincing. The authors measured the Euclidean distances instead of topological ones and Euclidean distances can be very misleading. This reviewer suggests the authors to calculate the topological distances (using Topolink or XWalk software) to make such comparison;
- 3) The data on the effect of chaotropic agents are also not firmly demonstrated. Following the cross-linking efficiency by SDS gel is not the best way to do it. Measuring the intact protein mass of the several species generated at each condition is a much more precise and reliable method.

Overall, is a nice manuscript and the cross-linking field will definitely improve with the data presented here but a more specific journal (like Anal. Chem.) should be more appropriated.

The comments from the reviewers are in blue and our responses are in black.

Reviewers' comments:

Reviewer #1 (Remarks to the Author):

A Non-hydrolyzable Lysine Cross-linker DOPA Enables Mass Spectrometry Analyses of Protein Unfolding and Weak Protein-protein Interactions

Jian-Hua Wang, Yu-Liang Tang, Rohit Jain, Fan Xiao, Zhou Gong, Yu Zhou, Dan Tan, Qiang Li, Xu Dong, C. Robert Matthews, Shu-Qun Liu, Chun Tang, Niu Huang, Keqiong Ye, Meng-Qiu Dong, Xiaoguang Lei

In this manuscript, the authors introduce a novel crosslinking reagent, di-ortho-phthalaldehyde crosslinker, DOPA2, which was shown to be non-hydrolyzable and amine-selective. In different experimental setups this reagent proved to have 6-10 times higher reaction rates than conventional NHS- esters like DSS, tolerated high concentrations of denaturing substances like Urea and guanidine hydrochloride and thus, was found to be predestined for monitoring the folding/unfolding process of proteins in short time scales and to detect weak and specific interactions.

Overall, I appreciate the development of new crosslinking reagents as they open up new avenues for different applications in the crosslinking field. However, in my opinion the added value of this reagent was insufficiently demonstrated and thus, I do not think that this work is suitable for publication in Nature Communications. I recommend to transfer this manuscript to a more specialized journal.

Major and Minor criticism:

1. I was irritated by the different numbers of how much faster the reaction rate of DOPA was in comparison to DSS: 6x, 10x, 60x?

We thank the reviewer for bringing up this important point. We have investigated this issue systematically. As summarized in Figure 1h, which is pasted below, the reaction rate difference between DOPA2 and DSS depends on the substrate type, i.e., whether the substrate is a protein, a peptide, or a free amino acid. With free lysine or peptide substrates, OPA is 3.5-7.7 times faster than the NHS ester. With protein substrates, DOPA2 is 60-120 times faster than DSS.

h

	Substrate = lysine or lysine-containing peptide			Estimation method	Substrate = protein		Rate of X-linking DOPA2/DSS
	OPA	NHS ester	OPA/NHS		DOPA2	DSS	
2 nd -order rate constant (M ⁻¹ s ⁻¹)	2.24*	0.36	6.2	BSA, X-linked dimer	10 s	10-20 min	60-120
2 nd -order rate constant (M ⁻¹ s ⁻¹) ^{1/2}	9.76	1.27	7.7	BSA, ~2000 X-linked spectra	10-20 s	20 min	60-120
The pseudo-first-order reaction rate constant (s ⁻¹)	3.18x10 ⁻⁴ *	9.10x10 ⁻⁵ #	3.5	Cameleon, 527/480 nm	< 20 s	> 20 min	> 60

*The second phase

#Assume 100% energy transfer in FRET

Figure 1. DOPA2 is a faster cross-linker than DSS.

(h) A summary for comparing the reaction rate of OPA and NHS ester, or DOPA2 and DSS.

Two FRET-based experiments were added to compare side-by-side the reaction rates of OPA and NHS ester, or DOPA2 and DSS, as detailed below.

(1) We performed a FRET assay to calculate the pseudo-first-order reaction rate constant of OPA and that of NHS ester on a peptide substrate (Figure 1a-c, Supplementary Figure 3).

Figure 1. DOPA2 is a faster cross-linker than DSS.

(a) FRET-based workflow of Cy3-K peptide reacting with Cy5-OPA or Cy5-NHS ester. (b) An example of the FRET signal change as the reaction between Cy3-K (12.5 μ M) and Cy5-NHS ester (25 μ M) proceeds. (c) FRET ratio as a function of reaction time. The reaction was initiated by mixing 25 μ M Cy3-K peptide with 2.5 μ M Cy5-OPA or Cy5-NHS ester.

Supplementary Figure 3. FRET-based system comparison of the reaction rate of Cy5-OPA and Cy5-NHS ester with Cy3-K peptide.

(a) The chemical structures of Cy3-K peptide, Cy5-OPA, and Cy5-NHS ester. (b) Fluorescence spectra of Cy3-K, Cy5-OPA, and Cy3-K (12.5 μ M) mixed with Cy5-OPA (25 μ M) for different amounts of time, subtracting the fluorescence intensity of Cy3-K.

(2) We performed a FRET assay to compare side-by-side the reaction speed of DOPA2 versus DSS on a protein substrate (Figure 1f-g).

Figure 1. DOPA2 is a faster cross-linker than DSS.

(f) FRET-based workflow of cross-linking cameleon calcium sensor protein (YC3.6) using DOPA2 or DSS. (CFP, cyan fluorescent protein; YFP, yellow fluorescent protein) (g) The emission fluorescence ratio of 527 and 480 nm with the increasing reaction time.

2. As mentioned in the discussion the DOPA reagent is sensitive to overcrosslinking in terms of reagent concentration and reaction time – this needed to be more carefully shown if you introduce a new reagent. What implications does overcrosslinking have?

Thank you for pointing it out. By over-cross-linking, we refer to a situation where many or most of the lysine side-chain amines are turned into either a mono-link or cross-link, that is, they can no longer be cut by trypsin. This leaves only arginine and the remaining unmodified lysine residues as cutting sites. Therefore, the resulting tryptic peptides tend to very long. The cross-linked long peptides are refractory to LCMS identification. We find that if a second protease Asp-N is used to digest such samples, the number of identified cross-linked peptide pairs increase by ~70% (Supplementary Figure 1d). Over-cross-linking can be avoided if we decrease the protein:DOPA2 ratio from 4:1 to 16:1 (w/w), in which case adding Asp-N is no longer necessary and trypsin alone is sufficient (Supplementary Figure 1d).

For DOPA cross-linking reactions, we tested different reaction time, pH, buffer systems, protein:cross-linker ratio (w/w), and temperature in a systematic manner (Figure 3a-b and Supplementary Figure 1). The optimal DOPA cross-linking conditions is specified in the Results section, on page 4, lines 142-145, “The highest number of cross-linked peptide pairs (referred to as cross-links for short) was obtained after a 10-minute reaction at 16:1 (BSA:DOPA, w/w), 25 °C, pH 7.4. Three buffer systems free of primary amines (HEPES, PBS, and trimethylamine) worked similarly well (Supplementary Figure 1a-c).”.

Supplementary Figure 1. Optimization of cross-linking conditions using BSA.

(a and b) Optimization of the working concentration of cross-linkers DOPA-C₂ and DOPA2. (c) Performance of DOPA2 in HEPES buffer, PBS buffer, and triethylamine buffer on BSA. (d) Comparison of cross-linking effects of DOPA2 by digestion with trypsin alone or with trypsin plus Asp-N.

Figure 3. Unique properties of DOPA2 compared with DSS.

(a) SDS-PAGE of DOPA2 or DSS-cross-linked BSA under different reaction times at 0 °C. (b) SDS-PAGE of DOPA2 or DSS-cross-linked BSA under different pH conditions.

3. In my opinion, the claim “monitoring the folding/refolding process of a protein” is overstating the capability of the method. As demonstrated it is able to identify residues or patches which are more or less sensitive to chaotropic agents, however, this does not necessarily describe the folding process.

We did not claim “monitoring the folding/refolding process of a protein”. We showed that DOPA2 cross-linking is an effective way to probe the unfolding process of RNase A. We have conducted additional experiments to demonstrate this type of application.

(1) DOPA2-assisted CXMS/XLMS/CL-MS succeeded in analyzing the unfolding process of RNase A in 3 M GdnHCl as well as in 6 M GdnHCl (Figure 6c-d).

(2) Computational modeling of the unfolding intermediates of RNase A (Figure 6f and Supplementary movie).

Figure 6. Analysis of the RNase A unfolding process by DOPA2 cross-linking.

(c) RNase A unfolding in 3 M GdnHCl. Heatmap representation of the abundance changes of the identified cross-linked residue pairs. Spectral counts were used to estimate abundance changes. (d) As in c, but in 6 M GdnHCl.

Figure 6. Analysis of the RNase A unfolding process by DOPA2 cross-linking.

(f) Four categories for simulated structures of RNase A in 8 M urea. Compared with the native structure, the lost secondary and tertiary structures are represented by dashed blue boxes and black boxes, respectively.

(3) DOPA2-assisted CXMS/XLMS/CL-MS succeeded in analyzing the unfolded states of

SNase in 8 M urea (Figure 4).

Figure 4. Analysis of the unfolded states of SNase by DOPA2 cross-linking.

(a) The crystal structure of SNase (PDB code: 1JOO). (b) The unfolding transition of SNase in different concentrations of urea monitored by fluorescence emission at 325 nm. (c) Circular dichroism spectra in far-UV region measured for SNase in the presence of urea. (d) Experimental workflow of cross-linking SNase in different concentrations of urea using DOPA2. (e) The changes of spectral counts for each identified cross-linked residue pair (normalized across the six conditions, shown as Z-scores on the left axis), and the mean residue ellipticity (MRE) of SNase monitored by CD at 222 nm (on the right axis) in different concentrations of urea. The residue pairs were classified into three clusters by K-means (Cluster A, Cluster B, and Cluster C). (f) The cross-links identified in different concentrations of urea were mapped on the crystal structure of RNase A (PDB code: 1JOO). Cluster A is framed in orange (23 pairs), Cluster B in blue (64 pairs), and Cluster C in green (7 pairs). A red line denotes that the distance between two cross-linked residues exceeds the maximal cross-linking distance of DOPA2, and a black denotes that it does not.

These experiments verified that DOPA2 is highly effective for characterization of protein unfolding intermediates and for capturing sequential conformational changes during protein unfolding.

Lastly, regarding the reviewer’s comment “it is able to identify residues or patches which are more or less sensitive to chaotropic agents, however, this does not necessarily describe the folding process,” we show that on peptide substrates, DOPA2 cross-linking is not sensitive to low pH or chaotropic agents (Figure 3 c and f). Then, what is the governing factor(s) behind DOPA2’s ability to identify residues or patches on proteins which are more or less sensitive to chaotropic agents? We have searched for the answer and the only one we can come up with is conformational change.

Figure 3. Unique properties of DOPA2 compared with DSS.

(c) The log₂ transformed MS1 peak intensity ratios of cross-linked products at pH 3.0 vs. pH 7.4. (f) The log₂ transformed MS1 peak intensity ratios of cross-linked products at GdnHCl (6 M) vs. a physiological buffer (HEPES, pH 7.4).

4. As I said before, the demonstration of the added value, of the new insights you can achieve with this reagents is not sufficient for Nature Communications.

Regarding the added value of the new cross-linking reagent, we show that DOPA cross-linking **provides insight into the process of protein conformational change**. This has impacts on two fronts: to revive a classic field of biology and to prepare for a new era of structural biology, as explained below.

The first one relates to protein folding/unfolding, which was ushered in by Anfinsen’s discovery of “amino acid sequence determines structure”, enriched by the discovery of chaperones, refined by computational modeling, but has stagnated since the 1990s due to a lack of tools to “provide insight into the folding/unfolding process” [PMID 30840846].

The second one has to do with the transforming power of AI technology. With the advent of AlphaFold 2 and RoseTTAFold, a sizable portion of the static structures of proteins can be predicted. We will soon be witnessing the frontier of structural biology moving away from static structures and advancing towards understanding protein dynamics, that is, from “snapshots” to “movies”. CXMS/XLMS/CL-MS can be a powerful tool in this movement, provided that the cross-linking reaction is fast enough to capture transient conformational changes. The existing

cross-linkers are all too slow for this purpose.

DOPA2 cross-linking, albeit only 60-120 times faster than NHS ester cross-linkers and thus can only probe slow conformational changes in time scales of seconds or minutes, is **a first step of next-generation CXMS**.

For comparison, we have summarized the reaction time of other cross-linkers in the following table, which is at least 15 min long. Cross-linking of proteins by DOPA2 can be as short as 10 seconds, even in urea or GdnHCl.

Cross-linker	Chemical structure	Targeted AA	Reaction time	Reference
DSS		Lys	30-60 min	Kim et al, 2018; Ryl et al, 2019; Ryl et al, 2020; Liu et al, 2020; Zhao et al, 2020.
BS ³		Lys	30-60 min	Yang et al, 2012; Kastritis et al, 2017; Linden et al, 2020.
DSSO		Lys	30-60 min	Kao et al, 2011; Liu et al, 2017; Yugandhar et al, 2019; O'Reilly et al, 2020.
DSBU		Lys	120 min (4 °C); 60 min (at RT)	Arlt et al, 2016; Pan et al, 2018; Ihling et al, 2019.
SDA		One end is Lys, and the other end is any amino acid.	incubation for 15-60 min and irradiation for 15-50 min	Belsom et al, 2017; Rega et al, 2018; Müller et al, 2019.
EDC		One end is Lys, and the other end is Asp/Glu.	60-120 min	Novak et al, 2008; Sriswasdi et al, 2014; Mintseris et al, 2019; Chang et al, 2021.

PDH/ADH		Asp/Glu	45-60 min	Leitner et al, 2014
Diazoker 1		Asp/Glu	60 min	Zhang et al, 2018
DHSO		Asp/Glu	60 min	Gutierrez et al, 2016
KArGO		One end is Lys, and the other end is Arg.	15 min	Jones et al, 2019

Reviewer #2 (Remarks to the Author):

This manuscript reports the use of di-ortho-phthalaldehyde (DOPA) as a cross-linker (XLer) for Lys and Ser residues as a replacement for NHS esters. The new cross-linking chemistry is well described and tested, including the effect of time, concentration, temperature, and the presence of chaotropic agents. The new XLer is tested against BSA, RNase A proteins and EIN/HPr and EIIA/EIIB complexes, besides a ten protein commercial mix. This is indeed an interesting contribution to the field of structural mass spectrometry.

My main issue with this manuscript is related to the relevance for the wider audience of Nature Communications when the following points:

1) Even though the data presented here are very interesting, it is an incremental improvement, i. e., the DOPA XLer behaves better than NHS esters but the gain are sometimes marginal or complementary in nature. I believe is a good contribution to the cross-linking field, but NHS esters can do most of what is presented here, just with a lower efficiency;

We are of the opinion that DOPA is more than an incremental improvement over the existing NHS ester cross-linkers. Please see our response to Reviewer 1's fourth concern.

2) One of the main attractiveness of DOPA is the higher structural compatibility of identified cross-links compared to NHS esters (II. Performance of DOPA vs DSS on model proteins) but

data is not convincing. The authors measured the Euclidean distances instead of topological ones and Euclidean distances can be very misleading. This reviewer suggests the authors to calculate the topological distances (using Topolink or XWalk software) to make such comparison;

We thank the reviewer for this helpful comment. We have calculated the topological distances and obtained the same conclusion. As shown in Figure 2e, the structural compatibility rate of DOPA2 cross-links (72.5%) or that of DOPA-C₂ (52.9%) is much higher than DSS cross-links (34.3%) by solvent accessible surface distance.

e

		DOPA-C ₂	DOPA2	DSS
Cross-linked residue pairs		138	197	422
Max. C α -C α distance (Å)		24.9*	30.2*	24.0
Structural compatibility (%)	Euclidean distance	78.99	88.32	55.21
	SASD	52.94	72.54	34.34

*Spacer arm w/t MM2 minimization

Figure 2. Evaluating the performance of DOPA.

(e) The table displays the percentage of residue pairs that are consistent with the structures of the model proteins, calculated by the use of the Euclidean distance or the solvent accessible surface distance.

3) The data on the effect of chaotropic agents are also not firmly demonstrated. Following the cross-linking efficiency by SDS gel is not the best way to do it. Measuring the intact protein mass of the several species generated at each condition is a much more precise and reliable method.

We appreciate the reviewer's concerns. In addition to SDS-PAGE, mass spec analysis was conducted to evaluate the effect of chaotropic agents (Figure 3f). More experimental evidence is added to the revised manuscript to show that DOPA2 is an effective and much needed tool to study protein unfolding induced by chaotropic agents:

- (1) DOPA2-assisted CXMS/XLMS/CL-MS succeeded in analyzing the unfolding process of RNase A in 3 M GdnHCl as well as in 6 M GdnHCl (Figure 6c-d).
- (2) Computational modeling of the unfolding intermediates of RNase A (Figure 6f and Supplementary movie)
- (3) DOPA2-assisted CXMS/XLMS/CL-MS succeeded in analyzing the unfolded states of SNase in 8 M urea (Figure 4).

Please see our response to Reviewer 1's third concern for the cited figures.

Overall, is a nice manuscript and the cross-linking field will definitely improve with the data presented here but a more specific journal (like Anal. Chem.) should be more appropriated.

Thank you for the kind words.

REVIEWER COMMENTS

Reviewer #1 (Remarks to the Author):

Fast amine-reactive, denaturant-compatible cross-linking with di-ortho-phthalaldehyde

(DOPA) uncovers protein unfolding intermediates

Jian-Hua Wang#, Yu-Liang Tang, Zhou Gong, Rohit Jain, Fan Xiao, Yu Zhou, Dan Tan, Qiang Li, Niu Huang, Shu-Qun Liu, Keqiong Ye, Chun Tang, Meng-Qiu Dong, Xiaoguang Lei

The authors describe here a novel crosslinking reagent, di-ortho-phthalaldehyde crosslinker, DOPA2, which was shown to be non-hydrolyzable and amine-selective. They demonstrated that DOPA2 has a 60-120 times faster rate of protein crosslinking compared to NHS esters like DSS and that DOPA2 is able to perform this faster crosslinking reactions in the presence of denaturing agents or at low pH or low temperatures. Using SNase and RNase A as a model substrate for monitoring a dynamic folding/unfolding process, they showed that distinct clusters of crosslinks indicate conformational changes during the unfolding process with increasing concentrations of denaturing agents. To some extent the abundance changes of crosslinks related the α -helical domains correlated with the changes indicated by CD analysis.

In this resubmitted manuscript the authors elaborate the characterization of the DOPA2 crosslinker agent with respect to reaction rate and insensitivity to denaturing agents in more detail. The initial characterization of the crosslinker thus corroborates the benefits of the o-phthalaldehyde based crosslinkers over the NHS ester chemistry.

However, I am still not convinced that the results obtained by applying DOPA2 to monitor the unfolding process of proteins justify publication of this manuscript in Nature communications.

Major comments:

1. A previous study (Walzthoeni et al., 2015) already applied the clustering of DSS crosslinks based on their abundance in order to describe the conformational states of the TRiC chaperonin. In this work the TRiC complex was stabilized in distinct functional and conformational states using different substrates and chemical agents. The crosslink clusters were then used to resolve the conformational differences between the states of this macromolecular protein complex.

This study needs to be cited in the manuscript.

2. Here, clusters of crosslinks indicate conformational changes in SNase or RNase A upon increasing urea/GdnHCl concentrations. These proteins are a lot smaller than the TRiC complex and thus the information content and also the resolution provided by the crosslink-derived distance restraints are too little to monitor changes of individual domains or secondary structures within these rather small proteins or at least it has not been demonstrated.

3. Although, the DOPA2 reagent is superior for protein crosslinking in denaturing conditions, I think that the examples picked in this study are not ideal for capturing the unfolding process of individual domains in a rather small protein by CXMS.

In addition, Walzthoeni et al. investigated distinct stabilized states of a macromolecular complex whereas this study attempts to capture a continuum of intermediates during the unfolding process which dramatically complicates the structural interpretation of crosslinks.

In my opinion, this study showed that DOPA2 crosslinking has the capability to indicate changes in the protein conformation under denaturing conditions, however, the structural insights were lacking coverage and resolution in order to describe the folding process.

Minor comments:

line 164/165: the rate constant is missing the M-1

line 201: structural compatibility: The significant difference between DOPA2 (88.3%) and DSS (55.2%) crosslinks that agree with 3D protein structures by applying Euclidean or SAS distances is rather a consequence of the different crosslinker structures and not of the crosslinker chemistries. DSS has an entirely aliphatic spacer arm whereas DOPA2 possesses the ethylene glycol spacer and the terminal ring structures of the ortho-phthalaldehyde. Thus, DOPA2 has a less flexible spacer in solution which might explain the smaller number of crosslink distances exceeding the span of the spacer arm.

Furthermore, the greater flexibility of the DSS spacer arm may also explain the higher number of crosslinks detected in the native conditions in comparison to DOPA2.

line 202: SASD is missing the figure reference

Reviewer #3 (Remarks to the Author):

The authors presented a nice crosslinker DOPA2 that allows extremely fast reaction in 10s. I can imagine DOPA2 may be used in many important biological scenarios where short crosslinking time is critical to capture prompt changes of protein interactions. I think the authors satisfactorily addressed the comments from reviewer2. However, since it's my first time to read this paper, I have two additional questions. The first one is about the authors' claim on the structural compatibility of DOPA2 crosslinker. In Figure 1e, they show that while increasing crosslinking time from 5s to 1h, spectra count increases from 1000 to 7000. 7000 seems a really high number for crosslinking of a single protein BSA. This makes me wonder if proteins undergo structural alternation or denaturation during crosslinking. To address this, I hope to see an experiment, ideally on a few protein complexes with known structures (could be from cell lysate or purified protein complexes), crosslink for different time duration, for instance 10s, 30s, 5min, and then measure structural compatibility. It is important to know if longer crosslinking time introduce more long-distance crosslinks, which I believe is a critical reference for future use of this crosslinker.

My second point is about the actual crosslinking time. While crosslinking takes 10s, quenching the reaction is 5min. Does it mean the actual crosslinking time is 5min plus 10s, or the 5min is only to make sure complete quenching? What is the quenching kinetics?

The comments from the reviewers are in blue and our responses are in black.

REVIEWER COMMENTS

Reviewer #1 (Remarks to the Author):

Fast amine-reactive, denaturant-compatible cross-linking with di-ortho-phthalaldehyde (DOPA) uncovers protein unfolding intermediates

Jian-Hua Wang#, Yu-Liang Tang, Zhou Gong, Rohit Jain, Fan Xiao, Yu Zhou, Dan Tan, Qiang Li, Niu Huang, Shu-Qun Liu, Keqiong Ye, Chun Tang, Meng-Qiu Dong, Xiaoguang Lei

The authors describe here a novel crosslinking reagent, di-ortho-phthalaldehyde crosslinker, DOPA2, which was shown to be non-hydrolyzable and amine-selective. They demonstrated that DOPA2 has a 60-120 times faster rate of protein crosslinking compared to NHS esters like DSS and that DOPA2 is able to perform this faster crosslinking reactions in the presence of denaturing agents or at low pH or low temperatures. Using SNase and RNase A as a model substrate for monitoring a dynamic folding/unfolding process, they showed that distinct clusters of crosslinks indicate conformational changes during the unfolding process with increasing concentrations of denaturing agents. To some extent the abundance changes of crosslinks related the α -helical domains correlated with the changes indicated by CD analysis.

In this resubmitted manuscript the authors elaborate the characterization of the DOPA2 crosslinker agent with respect to reaction rate and insensitivity to denaturing agents in more detail. The initial characterization of the crosslinker thus corroborates the benefits of the o-phthalaldehyde based crosslinkers over the NHS ester chemistry.

However, I am still not convinced that the results obtained by applying DOPA2 to monitor the unfolding process of proteins justify publication of this manuscript in Nature communications.

Major comments:

1. A previous study (Walzthoeni et al., 2015) already applied the clustering of DSS crosslinks based on their abundance in order to describe the conformational states of the TRiC chaperonin. In this work the TRiC complex was stabilized in distinct functional and conformational states using different substrates and chemical agents. The crosslink clusters were then used to resolve the conformational differences between the states of this macromolecular protein complex.

This study needs to be cited in the manuscript.

We have cited this study (Walzthoeni et al., 2015) in the revised manuscript.

2. Here, clusters of crosslinks indicate conformational changes in SNase or RNase A upon increasing urea/GdnHCl concentrations. These proteins are a lot smaller than the TRiC complex

and thus the information content and also the resolution provided by the crosslink-derived distance restraints are too little to monitor changes of individual domains or secondary structures within these rather small proteins or at least it has not been demonstrated.

Following the reviewer's advice, we have added in the revised manuscript DOPA2 cross-linking analysis of BSA, which consists of three domains and exists in a monomer-dimer equilibrium. The results are displayed in Figure 5 and Supplementary Figure 11. The most interesting findings are: (1) in a descending order of sensitivity to denaturation by urea are between-protein cross-linking, between-domain cross-linking, and within-domain cross-linking; (2) for cross-links that are more abundant in the denatured state than the native state (the non-native clusters), the two-linked lysine residues tend to be close in the primary sequence ($71\% < 50$ aa); (3) the cross-links of the non-native clusters are concentrated within individual domains, especially Domain III.

For SNase, which is a small, single-domain protein, DOPA2 cross-linking revealed that a subset of cross-links (Cluster C, Figure 4e), all within the alpha-subdomain and involving α -helix 3, are more sensitive to urea denaturation than the other ones identified from the native-state of SNase (Cluster A, Figure 4e). This suggests that α -helix 3 is unpacked before the breakdown of the rest of the domain.

These data demonstrate the utility of DOPA2 cross-linking in protein unfolding studies.

The text and the figures describing the newly added BSA result are copied below.

“Analyzing the unfolded states of BSA by DOPA2 cross-linking

Using the same method, we analyzed denaturation of BSA. This 583-aa protein is much larger than SNase and exists in a monomer-dimer equilibrium in solution. Each BSA monomer has three helical domains arranged in the shape of a heart: Domain I (1-172 aa) and Domain III (373-583 aa) are the left and the right atrium, respectively, joined by Domain II (173-372 aa), the left and right ventricles in one⁵¹.

Previous studies using a variety of techniques have shown that BSA follows a two-stage, three-state unfolding route going from zero to 8 M urea⁵²⁻⁵⁵. For example, both the CD signal in the far-UV region and the fluorescence intensity change of tryptophan residues, primarily Trp214 (Trp213 in Figure 5), report no or little change of BSA in 1 and 2 M urea, followed by a gradual increase or decrease up to 7 M urea, and no further change in 8 M urea⁵⁵. Though appearing to be a two-state transition, the tryptophan fluorescence exhibits blue shift going from zero to 4 M urea, and a red shift going from 5 to 8 M urea⁵⁵, which indicates the presence of an intermediate unfolding state in 4-5 M urea.

In this study, a total of 87 high-quality cross-links identified from the BSA samples in 0-8 M urea (Figure 5) characterized BSA denaturation in greater detail. As shown in Figure 5a and Supplementary Figure 8, the Lys524-Lys524 cross-link was the most sensitive one to perturbation by urea. It is the only inter-molecular cross-link identified unambiguously in this

experiment. Its abundance nearly halved upon 1 M urea and dropped all the way down upon 2 M urea.

The 16 cross-linked lysine pairs of Cluster 1 were the second most sensitive to urea (Figure 5b). The two linked residues in each pair tend to be far apart in the primary sequence (69% > 100 aa, 38% > 200 aa). In fact, seven of them are between-domain cross-links. In addition, we observed ten cross-links that bridge across the central cleft of the heart-shaped BSA monomer where there is a dearth of hydrogen bonds. The lack of a force to hold the cleft in position may explain why the cross-cleft cross-linking is easily disrupted by urea.

Less sensitive to urea were the cross-links of Cluster 2 (Figure 5c). Cluster 1 and 2 both diminished in abundance as the urea concentration increased. However, unlike Cluster 1, Cluster 2 cross-links were hardly affected up till 4 M urea, and all except three were within-domain cross-links.

Clusters 3-5 were the opposite of Clusters 1 and 2 (Figure 5d-f). As the concentration of urea increased to 8 M, cross-links of Cluster 3-5 either gradually or eventually but abruptly became the dominant species. Many of them are over-length cross-links if mapped to the native structure of BSA (marked by red line, Figure 5e-f), and the two-linked residues are typically not far apart in the primary sequence (98% < 100 aa, 71% < 50 aa).

Of the cross-links described above, Cluster 2 and Cluster 4, each showing a sharp transition between 4 M and 6 M urea, highlight the main unfolding phase of BSA and match up the reported intermediate unfolding states of BSA. The rest of the cross-links, including four unclassified cross-links (Supplementary Figure 9), bring details not accessed by other techniques.”

Figure 5. Analysis of the unfolded states of BSA by DOPA2 cross-linking.

(a) The spectral counts of the inter-molecular cross-link (Lys524-Lys524) are normalized as Z-scores and plotted against urea concentration. This cross-link is marked on the crystal structure of a BSA dimer (PDB code: 3V03), with Trp213 represented by purple sticks in one of the subunit. (b-f) As in (a), but for cross-links of Clusters 1-5. The cross-links are indicated on the structure of the BSA monomer, with Domain I, II, and III in light orange, green, and cyan, respectively. The color of a two lysine residues (highlighted in red) denotes whether or not the indicated cross-link fits within the maximal cross-linking distance of DOPA2 when mapped to the native structure of BSA (black, ≤ 30.2 Å; red, ≥ 30.2 Å). We performed two independent cross-linking experiments for each sample, and each was analyzed twice by LC-MS/MS. Cross-linking residue pairs were filtered by requiring FDR < 0.01 at the spectra level, E-value < 1×10^{-8} and spectral counts > 3.

Supplementary Figure 8. Annotated MS/MS spectrum of the DOPA2-linked inter-molecular peptide pair from BSA.

Supplementary Figure 9. The four unclassified cross-links of BSA by DOPA2 cross-linking in urea.

(a) Z-scores reporting the normalized spectral counts of four unclassified cross-links as a function of urea concentration. (b) The cross-links are indicated on the crystal structure of BSA (PDB code: 3V03).

3. Although, the DOPA2 reagent is superior for protein crosslinking in denaturing conditions, I think that the examples picked in this study are not ideal for capturing the unfolding process of individual domains in a rather small protein by CXMS.

In addition, Walzthoeni et al. investigated distinct stabilized states of a macromolecular complex whereas this study attempts to capture a continuum of intermediates during the unfolding process which dramatically complicates the structural interpretation of crosslinks.

In my opinion, this study showed that DOPA2 crosslinking has the capability to indicate changes in the protein conformation under denaturing conditions, however, the structural insights were lacking coverage and resolution in order to describe the folding process.

As described above, we have added another example (BSA) to demonstrate that DOPA2 cross-linking provides valuable insights to protein unfolding.

With BSA, SNase and RNase A, we show experimentally that DOPA2 cross-linking generate different sets of cross-links, and they provide structural sights to the unfolding process of each protein. Specifically, cross-links that arise from native states of a protein can disappear either gradually (e.g. Cluster 1 of BSA, Figure 5b and Supplementary Figure 11a) or abruptly (e.g. Cluster 2 of BSA, Figure 5c and Supplementary Figure 11b), in the presence of relatively low concentration of urea (e.g. Cluster C of SNase, Figure 4e) or higher (e.g. Cluster A of SNase, Figure 4e). Conversely, new cross-links appear as the concentration of urea increases and they correspond to the unfolded states of proteins. During the revision of the manuscript, we noticed that these cross-links appear at different paces (Figure 5d-f), indicating progressive unfolding of BSA.

Minor comments:

line 164/165: the rate constant is missing the M-1

Here we report the pseudo-first-order reaction rate constant, thus the unit is s^{-1} .

line 201: structural compatibility: The significant difference between DOPA2 (88.3%) and DSS

(55.2%) crosslinks that agree with 3D protein structures by applying Euclidean or SAS distances is rather a consequence of the different crosslinker structures and not of the crosslinker chemistries. DSS has an entirely aliphatic spacer arm whereas DOPA2 possesses the ethylene glycol spacer and the terminal ring structures of the ortho-phthalaldehyde. Thus, DOPA2 has a less flexible spacer in solution which might explain the smaller number of crosslink distances exceeding the span of the spacer arm.

Furthermore, the greater flexibility of the DSS spacer arm may also explain the higher number of crosslinks detected in the native conditions in comparison to DOPA2.

Our conclusion is that the higher structural compatibility rates of DOPA2 cross-links over DSS cross-links cannot be explained fully by the longer spacer arm of DOPA2. We agree with the reviewer that flexibility and hydrophobicity of the linker, and the terminal ring structures could all play a part. The revised sentence is: “Of note, the maximum allowed cross-linking distance of DOPA-C2 (24.9 Å) and that of DSS (24.0 Å) are similar, therefore the length of a cross-linker is unlikely the only determinant of structural compatibility; other chemical properties such as flexibility, hydrophobicity, and bulkiness of the cross-linker likely play a role.”

line 202: SASD is missing the figure reference

We have made the corresponding revisions in the revised manuscript.

Reviewer #3 (Remarks to the Author):

The authors presented a nice crosslinker DOPA2 that allows extremely fast reaction in 10s. I can imagine DOPA2 may be used in many important biological scenarios where short crosslinking time is critical to capture prompt changes of protein interactions. I think the authors satisfactorily addressed the comments from reviewer2. However, since it's my first time to read this paper, I have two additional questions. The first one is about the authors' claim on the structural compatibility of DOPA2 crosslinker. In Figure 1e, they show that while increasing crosslinking time from 5s to 1h, spectra count increases from 1000 to 7000. 7000 seems a really high number for crosslinking of a single protein BSA. This makes me wonder if proteins undergo structural alternation or denaturation during crosslinking. To address this, I hope to see an experiment, ideally on a few protein complexes with known structures (could be from cell lysate or purified protein complexes), crosslink for different time duration, for instance 10s, 30s, 5min, and then measure structural compatibility. It is important to know if longer crosslinking time introduce more long-distance crosslinks, which I believe is a critical reference for future use of this crosslinker.

We analyzed the cross-links identified after BSA or a heterodimeric protein complex PUD-1/2 was incubated with DOPA2 for different amount of time, from as short as 10 seconds to as long as 10 minutes, or more (Supplementary Figure 7). As the cross-linking time increased, more DOPA2 cross-linked peptides were identified (spectral counts and residue pairs both increased) (Supplementary Figure 7a and c). At ten seconds, the structural compatibility rate of the

identified cross-links was the highest (100 %) and remained above 80% with prolonged cross-link reactions (Supplementary Figure 7b and d by Euclidean distance). Of the cross-links identified from short cross-linking reactions, 90-100% were identified from longer reactions (Supplementary Figure 7e-f). Fewer DSS cross-links were identified in BSA and complex PUD-1/2 (Supplementary Figure 7g-l), especially in 10-second reactions (47 DOPA2 cross-links vs 14 DSS cross-links in BSA, 25 DOPA2 cross-links vs 0 DSS cross-links in PUD-1/2).

Supplementary Figure 7. Comparing DOPA2 or DSS cross-linking reactions of different durations.

(a) Cross-links identified in BSA after cross-linking by DOPA2 for the indicated amount of time. The number of cross-link spectra is represented by blue columns, and the number of cross-linked residue pairs by orange columns. (b) The fraction of residue pairs that are consistent with the structure of BSA (PDB code: 3V03), as calculated by the use of the Euclidean distance or the solvent accessible surface distance (SASD). (c and d) As in a-b, but for PUD-1/2 complex (PDB code: 4JDE). (e) Venn diagram showing the overlap of DOPA2 cross-links identified in BSA in short reactions and those in longer reactions. The cross-links identified from 10-, 20-, 30-, and 40-s reactions were combined into Group A and the cross-links identified from 1-, 2-, 3-, and 10-min reactions were combined into Group B. (f) As in (e), but for PUD-1/2 complex. (g-l) As in a-f, but cross-linked by DSS. Cross-links were filtered by requiring FDR < 0.01 at the spectra level, E-value < 1×10^{-3} .

My second point is about the actual crosslinking time. While crosslinking takes 10s, quenching the reaction is 5min. Does it mean the actual crosslinking time is 5min plus 10s, or the 5min is only to make sure complete quenching? What is the quenching kinetics?

Thank you so much for bringing up this point! Fast quenching is an important element of the DOPA2 workflow that we forgot to discuss in the last submitted manuscript. We have now included the relevant data (Supplementary Figure 6). As shown, ~90% of the DOPA2 mono-links that retain one reactive end were blocked by hydrazine within 5 seconds. The

completeness of quenching by hydrazide remained at ~90% with longer incubation time. We conclude that DOPA2 cross-linking time can be controlled precisely: by quenching the reaction with hydrazine, cross-linking essentially stops within 5 seconds.

The related test and figure are copied below.

“Fast cross-linking can potentially be used to capture dynamic changes of proteins, if the reaction can be stopped promptly. We therefore screened a panel of compounds (ammonia, methylamine, hydrazine, methoxyammonium chloride, and *o*-benzylhydroxylamine) and found that hydrazine can quench OPA in 5 seconds (Supplementary Figure 6). It is possible that hydrazine may quench the reaction in < 5 seconds, but with manual pipetting we were unable to test shorter time points.”

Supplementary Figure 6. DOPA2 cross-linking of BSA was quenched by hydrazine within five seconds.

Ten seconds after DOPA2 (0.17 mM, final conc.) was added to a 1mg/mL BSA solution, hydrazine (20 mM, final conc.) was added to the mixture and let incubate for 5 s, 10 s, ... and up to 10 min. Then, six volumes of cool acetone was added to precipitate BSA. After the hydrazine-containing supernatant was quickly removed, the pellet was digested with trypsin and analyzed by LCMS.

(a) Illustration of the structures of mono-linked peptides before (mLK1) and after (mLK2) quenching by hydrazine.
 (b) Percentage of spectral counts of mLK1 and that of mLK2 after the DOPA2-BSA reaction was quenched by hydrazine for the indicated amount of time. The number of spectra identified is shown on the bar. The data was filtered by requiring FDR < 1% at the spectra level.

REVIEWERS' COMMENTS

Reviewer #3 (Remarks to the Author):

The authors satisfactorily addressed my concerns therefore I support the publication of the manuscript.